# Towards Global Optimality in Cooperative MARL with Sequential Transformation

## Abstract

Policy learning in multi-agent reinforcement learning (MARL) is challenging due to the exponential growth of joint state-action space with respect to the number of agents. To achieve higher scalability, the paradigm of centralized training with decentralized execution (CTDE) is broadly adopted with factorized structure in MARL. However, we observe that existing CTDE algorithms in cooperative MARL cannot achieve optimality even in simple matrix games. To understand this phenomenon, we analyze two mainstream classes of CTDE algorithms – actor-critic algorithms and value-decomposition algorithms. Our theoretical and experimental results characterize the weakness of these two classes of algorithms when the optimization method is taken into consideration, which indicates that the currently used centralized training manner is deficient in compatibility with decentralized policy. To address this issue, we present a transformation framework that reformulates a multi-agent MDP as a special "single-agent" MDP with a sequential structure and can allow employing off-the-shelf single-agent reinforcement learning (SARL) algorithms to efficiently learn corresponding multi-agent tasks. After that, a decentralized policy can still be learned by distilling the "single-agent" policy. This framework retains the optimality guarantee of SARL algorithms into cooperative MARL. To instantiate this transformation framework, we propose a Transformed PPO, called T-PPO, which can theoretically perform optimal policy learning in the finite multi-agent MDPs and shows significant outperformance on a large set of cooperative multi-agent tasks.

## 1 Introduction

Cooperative multi-agent reinforcement learning (MARL) is a promising approach to a variety of real-world applications, such as sensor networks (Zhang & Lesser, 2011), traffic light control (Van der Pol & Oliehoek, 2016), and multi-robot formation (Alonso-Mora et al., 2017). However, "the curse of dimensionality" is one major challenge in cooperative MARL, since the joint state-action space grows exponentially with respect to the number of agents. To achieve higher scalability, the paradigm of centralized training with decentralized execution (CTDE) (Kraemer & Banerjee, 2016a) is wildly used, which allows agents to learn their local policies in a centralized way while retaining the ability of decentralized execution.

Recently, many CTDE algorithms have been proposed. For value-based methods, the joint Q value is factorized as a function of individual Q values of agents (for which they are also called value-decomposition algorithms), and then standard TD-learning is applied. To enable scalability and decentralized execution, it is critical to ensure the joint greedy action can be computed by selecting local greedy actions through individual Q functions, which is formalized as the Individual-Global-Max (IGM) principle (Son et al., 2019). Based on this IGM property, a series of factorized multi-agent Q-learning methods have been developed, including but not limited to VDN (Sunehag et al., 2018), QMIX (Rashid et al., 2018), QTRAN (Son et al., 2019), and QPLEX (Wang et al., 2021b). For multi-agent actor-critic methods, the joint policy is often factorized into the direct product of individual policies, each of which is learned through policy gradient updates. For example, COMA (Foerster et al., 2018) and DOP (Wang et al., 2021c) aim at the critic design for effective credit assignment, MADDPG (Lowe et al., 2017) studies the situation with parameterized deterministic policies, and MAPPO (Yu et al., 2021) applies PPO to multi-agent settings with parameter sharing.

Despite the promising performance in benchmark tasks, these CTDE methods do not have a global optimality guarantee in general cooperative settings and may fail to achieve optimality even in simple matrix games (Section 3). It might be confusing since some algorithms like QPLEX (Wang et al., 2021b) have been proven to converge to the global optimum in some theoretical work (Wang et al., 2021a), which contradicts our experimental results. To understand this phenomenon, we provide theoretical analysis for both actor-critic algorithms and value-decomposition algorithms. It shows that when the optimization method is taken into consideration, which prior analysis didn't, neither actor-critic algorithms nor value-decomposition algorithms can get out of local optimums, yet which wildly exists in multi-agent tasks (Section 3).

To address this suboptimality issue, we present a transformation framework that reformulates a multi-agent MDP as a special "single-agent" MDP with a sequential decision-making structure among agents. With this transformation, any off-the-shelf single-agent reinforcement learning (SARL) method can be adopted to efficiently learn coordination policies in cooperative multi-agent tasks by solving the transformed single-agent tasks with their global optimality guarantee retained. To enable decentralized execution, a decentralized policy is learned at the same time by distilling the "single-agent" policy. As an instantiation of this transformation framework, a Transformed PPO (T-PPO) is proposed, which can theoretically perform optimal policy learning in finite-multi-agent MDPs and empirically shows significant outperformance on a large set of cooperative multi-agent tasks, including SMAC (Samvelyan et al., 2019) and GRF (Kurach et al., 2019) using attention mechanism (Vaswani et al., 2017).

## 2 PRELIMINARIES & RELATED WORK

### 2.1 RL MODELS

In single-agent RL (SARL), an agent interacts with a Markov Decision Process (MDP) to maximize its cumulative reward (Sutton & Barto, 2018). An MDP is defined as a tuple $(\mathcal{S}, \mathcal{A}, r, P, \gamma, s_0)$, where $\mathcal{S}$ and $\mathcal{A}$ denote the state space and action space. At each time step $t$, the agent observes the state $s_t$ and chooses an action $a_t \in \mathcal{A}$, where $a_t \sim \pi(s_t)$ depends on $s_t$ and its policy $\pi$. After that, the agent will gain an instant reward $r_t = r(s_t, a_t)$, and transit to the next state $s_{t+1} \sim P(\cdot|s_t, a_t)$. $\gamma$ is the discount factor. The goal of an SARL agent is to optimize a policy $\pi$ that maximizes the expected cumulative reward, i.e., $\mathcal{J}(\pi) = \mathbb{E}_{s_{t+1} \sim P(\cdot|s_t, \pi(s_t))} \left[ \sum_{t=0}^{\infty} \gamma^t r(s_t, \pi(s_t)) \right]$.

In MARL, we model a fully cooperative multi-agent task as a Dec-POMDP (Oliehoek et al., 2016) defined by a tuple $\langle \mathcal{N}, \mathcal{S}, \mathcal{A}, P, \Omega, O, r, \gamma \rangle$, where $\mathcal{N}$ is a set of agents and $\mathcal{S}$ is the global state space, $\mathcal{A}$ is the action space, and $\gamma$ is a discount factor. At each time step, agent $i \in \mathcal{N}$ has access to the observation $o_i \in \Omega$, drawn from the observation function $O(s, i)$. Each agent has an action-observation history $\tau_i \in \Omega \times (\mathcal{A} \times \Omega)^*$ and constructs its individual policy $\pi(a_i|\tau_i)$. With each agent $i$ selecting an action $a_i \in \mathcal{A}$, the joint action $\boldsymbol{a} \equiv [a_i]_{i=1}^n$ leads to a shared reward $r = R(s, \boldsymbol{a})$ and the next state $s'$ according to the transition distribution $P(s'|s, \boldsymbol{a})$. The formal objective of MARL agents is to find a joint policy $\boldsymbol{\pi} = \langle \pi_1, \ldots, \pi_n \rangle$ conditioned on the joint trajectories $\boldsymbol{\tau} \equiv [\tau_i]_{i=1}^n$ that maximizes a joint value function $V^{\boldsymbol{\pi}}(s) = \mathbb{E} \left[ \sum_{t=0}^{\infty} \gamma^t r_t | s_0 = s, \boldsymbol{\pi} \right]$. Another quantity in policy search is the joint action-value function $Q^{\boldsymbol{\pi}}(s, \boldsymbol{a}) = r(s, \boldsymbol{a}) + \gamma \mathbb{E}_{s'}[V^{\boldsymbol{\pi}}(s')]$.

To simplify our analysis, we present a framework of Multi-agent MDPs (MMDP) (Boutilier, 1996), a special case of Dec-POMDP, to model cooperative multi-agent decision-making tasks with full observations. MMDP is defined as a tuple $\langle \mathcal{N}, \mathcal{S}, \mathcal{A}, P, r, \gamma \rangle$, where $\mathcal{N}, \mathcal{S}, \mathcal{A}, P, r$, and $\gamma$ are the same agent set, state space, action space, transition function, reward function, and discount factor in Dec-POMDP, respectively. Due to the full observations, at each time step, the current state $s$ is observable to each agent. For each agent $i$, an individual policy $\hat{\pi}_i(a|s)$ represents a distribution over actions conditioned on the state $s$. Agents aim to find a joint policy $\hat{\boldsymbol{\pi}} = \langle \hat{\pi}_1, \ldots, \hat{\pi}_n \rangle$ that maximizes a joint value function $\widehat{V}^{\hat{\boldsymbol{\pi}}}(s)$, where denoting $\widehat{V}^{\hat{\boldsymbol{\pi}}}(s) = \mathbb{E} \left[ \sum_{t=0}^{\infty} \gamma^t r_t | s_0 = s, \hat{\boldsymbol{\pi}} \right]$.

### 2.2 POLICY FACTORIZATION AND CENTRALIZED TRAINING WITH DECENTRALIZED EXECUTION

In MARL, due to partial observability and communication constraints, a decentralized policy is required during execution, i.e., the joint execution policy $\boldsymbol{\pi}^{test}$ can be decomposed into a product of

individual execution policies $[\pi_i^{test}]_{i=1}^n$, called *policy factorization*:

$$\forall \boldsymbol{\tau}: \quad \boldsymbol{\pi}^{test}(\boldsymbol{a}|\boldsymbol{\tau}) = \prod_{i=1}^n \pi_i^{test}(a_i|\tau_i). \tag{1}$$

In order to effectively learn $\boldsymbol{\pi}^{test}$, centralized training with decentralized execution (CTDE) becomes a popular paradigm of cooperative MARL (Oliehoek et al., 2008; Kraemer & Banerjee, 2016b). In CTDE, agents are trained in a centralized manner and are granted access to other agents' information or global state during the centralized training process. Denote the joint policy which is learned during training as $\boldsymbol{\pi}^{train}$. Note that during centralized training, the constraint of policy factorization (see Eq. (1)) is not necessary for $\boldsymbol{\pi}^{train}$ and the agents can use joint policy $\boldsymbol{\pi}^{train}$ to interact with the environments for collecting training data. However, most popular CTDE MARL algorithms (Foerster et al., 2018; Lowe et al., 2017; Wang et al., 2021c; Yu et al., 2021; de Witt et al., 2020; Sunehag et al., 2018; Rashid et al., 2018; Son et al., 2019; Wang et al., 2021b) encode the policy factorization defined in Eq. (1) into the training joint policy $\boldsymbol{\pi}^{train}$ and these algorithms can be divided into two categories: actor-critic and value-decomposition.

For multi-agent actor-critic algorithms (Foerster et al., 2018; Lowe et al., 2017; Wang et al., 2021c; Yu et al., 2021; de Witt et al., 2020), a joint (stochastic or deterministic) policy is represented as a product of individual policies, i.e., $\boldsymbol{\pi}(\boldsymbol{a}|\boldsymbol{\tau}) = \prod_{i=1}^N \pi_i(a_i|\tau_i)$, which corresponds to policy factorization defined in Eq. (1). After that, some estimation of multi-agent policy gradient (Kuba et al., 2021) is calculated through the critic to update the policy.

For value-decomposition algorithms (Sunehag et al., 2018; Rashid et al., 2018; Wang et al., 2021b), the joint Q value is decomposed as a function of local Q values with some parameter $\lambda \in \Lambda$ (Fu et al., 2022).

$$Q_{\text{jt}}(\boldsymbol{\tau}, \boldsymbol{a}) = f_{\text{mix}}(Q_1(\tau_1, \cdot), \cdots, Q_n(\tau_n, \cdot), \boldsymbol{\tau}, \boldsymbol{a}; \lambda) \tag{2}$$

After that, standard TD-learning is applied, and the IGM (*Individual-Global-Max*) principle (Son et al., 2019) is enforced to realize effective TD-learning, which asserts the consistency between joint and local greedy action selections in the joint action-value $Q_{tot}(\boldsymbol{\tau}, \boldsymbol{a})$ and local action-values $[Q_i(\tau_i, a_i)]_{i=1}^n$, respectively:

$$\forall \boldsymbol{\tau}: \quad \arg\max_{\boldsymbol{a} \in \mathcal{A}} Q_{tot}(\boldsymbol{\tau}, \boldsymbol{a}) = \left\langle \arg\max_{a_1 \in \mathcal{A}} Q_1(\tau_1, a_1), \ldots, \arg\max_{a_n \in \mathcal{A}} Q_n(\tau_n, a_n) \right\rangle. \tag{3}$$

Denote the greedy joint policy as $\boldsymbol{\pi}(\boldsymbol{a}|\boldsymbol{\tau}) = \arg\max_{\boldsymbol{a} \in \mathcal{A}} Q_{tot}(\boldsymbol{\tau}, \boldsymbol{a})$ and the greedy individual policies as $\pi_i(a_i|\tau_i) = \arg\max_{a_i \in \mathcal{A}} Q_i(\tau_i, a_i)$. We have $\boldsymbol{\pi}(\boldsymbol{a}|\boldsymbol{\tau}) = \prod_{i=1}^N \pi_i(a_i|\tau_i)$, which is called policy factorization defined in Eq. (1). Although various value factorizations (Wang et al., 2021a; Fu et al., 2022) are widely studied in the literature, to our best knowledge, this paper is the first to study the effect of CTDE from the perspective of optimal policy learning with optimization methods.

## 3 SUBOPTIMALITY OF CURRENT CTDE ALGORITHMS: INHERENT LOCAL-MINIMA STRUCTURE IN LOSS FUNCTION

In this section, we will formally analyze the suboptimality of multi-agent actor-critic and value-decomposition algorithms when taking the optimization method into consideration. In short, the manner of centralized training adopted by multi-agent actor-critic and value-decomposition methods creates inherent local-minima structures in their loss function, which makes gradient-descent methods lose optimality guarantee in general.

The main results of this section are summarized as Theorem 3.1 and Theorem 3.2. These two theorems elucidate the existence of local minima in the loss function of both actor-critic and value-decomposition algorithms. For multi-agent actor-critic algorithms, any Nash's equilibrium of policies always constitutes a stationary point of the loss function of actors, regardless of the parameterization of policy networks (e.g. sharing parameter (Cobbe et al., 2020) or not) and credit assignment in gradient calculation (Foerster et al., 2018), which implies multi-agent policy-gradient can reach

zero even when current policies are not optimal. Consequently, it makes multi-agent actor-critic algorithms lose optimality guarantee when gradient descent-based optimization methods are used. For value-decomposition algorithms, we will prove that the loss function of any value-decomposition method could have exponentially many local optima as long as the Q-function class is complete for the IGM condition. In the research of value-decomposition, great efforts have been made by a series of work (Sunehag et al., 2018; Rashid et al., 2018; Wang et al., 2021b) to enrich the Q-function class and make it complete since incomplete function would diverge the process of TD-learning (Wang et al., 2021a). However, our result shows that the richness of Q-function class and the smoothness of the loss function cannot retain at the same time under the IGM condition, which leads the gradient descent-based optimization methods to fail in this case.

**Actor-Critic algorithms**     On the one hand, for multi-agent actor-critic algorithms on MMDP, the actor represents a decentralized stochastic policy $\pi_{jt}(\boldsymbol{a}|s;\theta) = \prod \pi_i(a_i|s;\theta)$. And the actor loss is a primitive function of policy gradient, thus the policy is updated via multi-agent policy gradient (Leonardos et al., 2022). We can prove the following theorem:

**Theorem 3.1.** *For multi-agent actor-critic algorithms, any Nash's Equilibrium of policies is a stationary point of actor loss. Moreover, there exists a family of single step-MMDP such that the actor loss function contains $\Omega(|\mathcal{A}|)$ different local minimums for deterministic policy and infinite local minimums for stochastic policy.*

This theorem suggests that multi-agent actor-critic algorithms create inherent local minimums for their actor loss functions, and may get stuck in any Nash's equilibrium of policy, which can be arbitrarily worse than optimal policy (Table 1). The proof is presented in Appendix A.1.

Table 1: Matrix Game with multiple Nash Equilibria: 2 players (one selects a row and one selects a column) coordinate to select an entry of the matrix representing the joint payoff. In this case, $(0,0),(1,1),(2,2)$ are three different Nash's Equilibria with different payoffs $10, 5, 1$. Only $(0,0)$ is the globally optimal policy. However, policy $(1,1)$ and $(2,2)$ are two local minima of the actor loss, no matter what parameterizations of local policies are due to Theorem 3.1.

| 10 | -20 | -20 |
|----|-----|-----|
| -20 | 5 | 0 |
| -20 | 0 | 1 |

**Value-decomposition algorithms**     On the other hand, for a value-decomposition algorithm, the joint Q-value function is decomposed as Eq. (2), and updated by standard TD-loss. We are able to prove the following theorem:

**Theorem 3.2.** *There exists a family of MMDP, such that for any value-decomposition algorithm with a complete Q-function class satisfying the IGM condition (Eq. (3)), the TD-loss function contains $\Omega\left(|A|^{|\mathcal{S}|}\right)$ different local optima.*

The term "complete Q-function class" refers to that any function in $\mathbb{R}^{|\mathcal{S}| \times |\mathcal{A}|^n}$ is decomposable by the function $f_{\mathrm{mix}}$, like that in QPLEX (Wang et al., 2021b). This theorem suggests that, for value-decomposition methods with IGM condition satisfied, then either it has an incomplete function class (e.g. VDN (Sunehag et al., 2018), QMIX (Rashid et al., 2018)), or it's loss function contains exponentially many local optimums (e.g. QPLEX, it contains infinite local optimums (Proposition A.1), worse than what is claimed in Theorem 3.2), both of which loses optimality guarantee when gradient-based optimization method is adopted. The proof is presented in Appendix A.1.

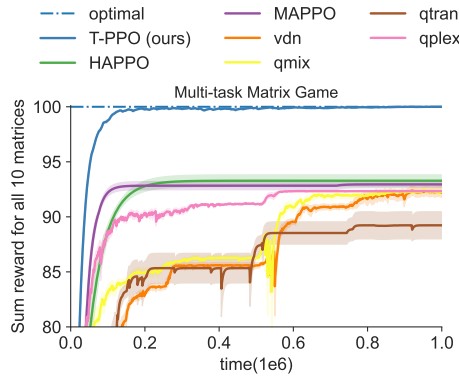

Figure 1: Learning curve of Multi-task Matrix Game

To give some experimental indication of our theorems, we introduce a multi-task matrix game here,

which is a simple 1-step game with 2 players and 10 matrices (see Appendix B.1.1 for details). We demonstrate the sum reward of all matrices for our approach and baselines in Figure 1. Baseline algorithms get stuck in locally optimal solutions, which confirms the theorems we discussed above. And with the theoretical guarantee of global optimality, our method (see Section 4.2) converges to 100 immediately. Value-based methods VDN, QMIX, and QPLEX converge to a similar local optimal point at the end of training. Taking benefit from the duplex dueling network architecture, QPLEX has a stronger representative ability than VDN and QMIX. Nevertheless, such a complex structure creates inherent local optima for its loss function (Theorem 3.2, Proposition A.1). QTRAN (Son et al., 2019) achieves convincing performance in some simple matrix games with a carefully designed regularizer tackling IGM. However, its discontinuous loss function will still hinder the globally optimal solution in this case (Appendix A.1). SOTA multi-agent actor-critic algorithms HAPPO (Kuba et al., 2021) and MAPPO get stuck in Nash's Equilibrium and cannot guarantee global optimality (Theorem 3.1). In this way, our approach dominates in our multi-task matrix game.

# 4 A SEQUENTIAL TRANSFORMATION FRAMEWORK FOR MARL PROBLEMS

To achieve global optimality in cooperative MARL with the CTDE paradigm, we try to design a new manner of centralized training to fit decentralized policy better with gradient descent-based optimization. Inspired by a series of prior works on sequence modeling (Bertsekas, 2019; Angermueller et al., 2020; Jain et al., 2022; Olivecrona et al., 2017; You et al., 2018), we found that breaking joint learning into sequential decision-making directly transfers properties of single-agent algorithms into multi-agent ones. To formalize such "reduction" of algorithms under the perspective of optimality guarantee, we propose a transformation framework that allows us to employ any single-agent RL (SARL) algorithm to efficiently learn corresponding multi-agent tasks with optimality guarantee of the SARL algorithm (if it has) kept.

From the theoretical perspective, we also find that this transformation framework keeps the same mini-max sample complexity (Appendix A.4). However, in order to maintain sample efficiency under this framework, one main challenge is to deal with the long horizon and sparse reward (Appendix A.4), while they have been long-term challenges for the design of empirical reinforcement learning algorithms (Arjona-Medina et al., 2018; Zheng et al., 2018; Gangwani et al., 2020). To tackle this challenge and instantiate our transformation framework, we propose a Transformed PPO, called T-PPO adopting the attention mechanism in network design, which incorporates the remarkable empirical ability of PPO (Schulman et al., 2017b) and the attention mechanism (Vaswani et al., 2017) as well as a theoretical optimality guarantee in the finite MMDPs. In this way, our implementation of T-PPO also bridges the gap between theoretical analysis and empirical performance.

## 4.1 SEQUENTIAL TRANSFORMATION

In the training phase of CTDE, all $n$ agents will coordinate to improve their joint policy. In particular, when a joint action needs to be inferred, they coordinate to infer $\boldsymbol{a} = (a_1, \cdots, a_n)$ jointly. If we give these agents a virtual order, imagine they infer their individual actions one after another, that is, in the centralized training phase, when agent $i$ infers its action $a_i$, all "previously inferred" actions $a_j (j < i)$ are known to $i$. As we shall see, this procedure of multi-agent decision-making is equivalent to a single-agent one since all agents are homogeneous in the centralized training phase. From a theoretical perspective, the discussion above states a transformation from MMDP into MDP in essence.

**Definition 4.1** (Sequential Transformation $\Gamma$, informal). *Given an MMDP $\mathcal{M}$ with $n$ agents, its sequential transformation is an MDP $\Gamma(\mathcal{M})$, that (1) for every time step $t \bmod n = 1$, the agent infers the action $a_1^{(\hat{t})}$ and transit from a state $s$ to a virtual state $(s, a_1^{(\hat{t})})$. (2) for every time step $t \bmod n = i > 1$, it infers an action $a_i^{(\hat{t})}$ and transit from virtual state $(s, a_{<i}^{(\hat{t})})$ to $(s, a_{\leq i}^{(\hat{t})})$. (3) for every time step $t \bmod n = 0$, it infers an action $a_n^{(\hat{t})}$ and transit from virtual state $(s, a_{<n}^{(\widehat{t})})$ to state $s'$ according to the dynamics of MMDP $\mathcal{M}$, at the same time, it gain a reward from $\mathcal{M}$. $\hat{t} = \lceil t/n \rceil$ here is the corresponding time step on $\mathcal{M}$.*

We present the formal definition in Definition A.1 for completeness. We can then prove the equivalence of the MMDP $\mathcal{M}$ and the MDP $\Gamma(\mathcal{M})$ after transformation in the perspective of policy value, which is presented in appendix (Theorem A.1) due to the lack of space. This theorem also provides the conversion method between policy on $\Gamma(\mathcal{M})$ and joint policy on $\mathcal{M}$.

This transformation allows us to use any SARL algorithm $A$, wrap the interface of the multi-agent environment to make $A$ run as if it is accessible to the interactive environment of $\Gamma(\mathcal{M})$, and finally convert the policy learned by $A$ to a joint policy on $\mathcal{M}$. The sequential framework converting SARL algorithm $A$ to MARL algorithm T-$A$ is formally described in pseudo-code in Appendix (Algorithm 1).

At last, we can claim the main theorem of this framework.

**Theorem 4.1** (The transformation framework keeps the optimality). *Using the sequential framework (Algorithm 1), if the SARL algorithm $\mathfrak{A}$ is guaranteed to obtain an optimal policy on MDP, then the MARL algorithm T-$\mathfrak{A}$ is guaranteed to obtain an optimal policy on MMDP.*

Moreover, thanks to the theoretical analysis for the global optimality of PPO with neural networks ((Liu et al., 2019)), we are able to claim a suitable implementation of T-PPO attains the global optimality under the same mild assumptions in Liu et al. (2019).

**Proposition 4.1** (Suitable implementation of T-PPO has optimality guarantee). *T-PPO converges to global optimum, if Assumption 4.1, 4.2, and 4.3 in Liu et al. (2019) hold, in particular, if $\mathcal{M}$ is tabular.*

The proofs of both propositions are presented in Section A.3, and more analysis of the complexity of the transformed model is presented in Section A.4.

## 4.2  T-PPO: EXTENSION TO DEC-POMDP

In practice, many MARL benchmarks are partially observable. To apply our algorithms to partial-observable environments, we need to extend our algorithms to Dec-POMDP. To instantiate our transformation framework, we propose a Transformed PPO (T-PPO) based on PPO (Schulman et al., 2017a). The Actor-Critic structure is shown in Figure 2. Following the sequential transformation framework discussed in Section 4.1, we introduce previous agents' actions to each agent's actor and critic modules. However, this sequential transformed information increases with respect to the number of agents. To achieve scalability, we equip each agent's actor and critic with a multi-head attention (MHA) module.

Intuitively, we believe considering too much information from previous agents' actions is harmful to learning since in most cases, we do not need that much information for a single agent to make a decision. So we add regularization terms to agents' actor and critic modules to encourage each agent to extract critical information. As demonstrated by yellow modules in actor, policies $\pi_{i,T}^{(t)}$ used for training is combined with two parts: one part contains previous agents' actions as inputs for the MHA module, and the other part only takes the individual trajectory as input ($\pi_{i,main}^{(t)}$). For regularization, we add KL divergence between $\pi_{i,main}^{(t)}$ and $\pi_{i,T}^{(t)}$ to decrease the influence of previous agents' actions. A similar structure is also implemented in the critic structure, as shown on the right side of Figure 2, with L1 norm for regularization. To enable decentralized execution, we further distill the single-agent policy $\pi_{i,T}^{(t)}$ to a decentralized policy $\pi_{i,E}^{(t)}$ with behavior cloning by optimizing the cross entropy loss independently for each agent, which is equivalent to minimizing the KL-divergence between the joint policy and the joint decentralized policy (see Appendix B.2.3 for detail). Here we share GRU and the representation layer, whose inputs do not contain other agents' information.

## 5  EXPERIMENTS

We design experiments to answer the following questions: (1) Can the proposed sequential framework achieve the globally optimal policy on MMDP? (Section 3 and Section 5.1.1) (2) Can our approach improve learning efficiency for policy-based MARL algorithms? (Section 5.1 and Section

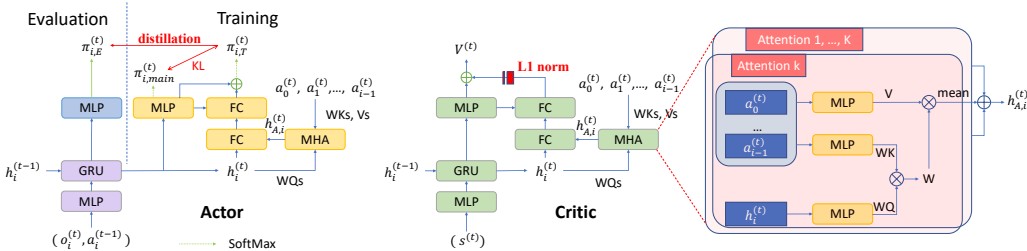

Figure 2: The architecture of combining our sequential transformation framework with PPO (T-PPO)

5.2) (3) Will distillation influence performance during decentralized execution? (Section 5.1 and Section 5.2)

We first use a multi-task matrix game to demonstrate the global optimality of our single-agent policy compared to multi-agent value-based methods (VDN, QMIX, QTRAN, QPLEX) and policy-based methods (MAPPO, HAPPO(Kuba et al., 2021)). This part has been discussed in Section 3. Then, we use challenging tasks from the StarCraft II micromanagement (SMAC) benchmark(Samvelyan et al., 2019) and Google Research Football (GRF) benchmark (Kurach et al., 2019) to further demonstrate and illustrate the outperformance of our approach. In each environment, We show the average and variance of the performance for our method and baselines tested with three random seeds (seed 0, 1, 2). For all baselines, we use the codes provided by the authors properly with the same hyper-parameters as the original papers.

## 5.1 STARCRAFT II

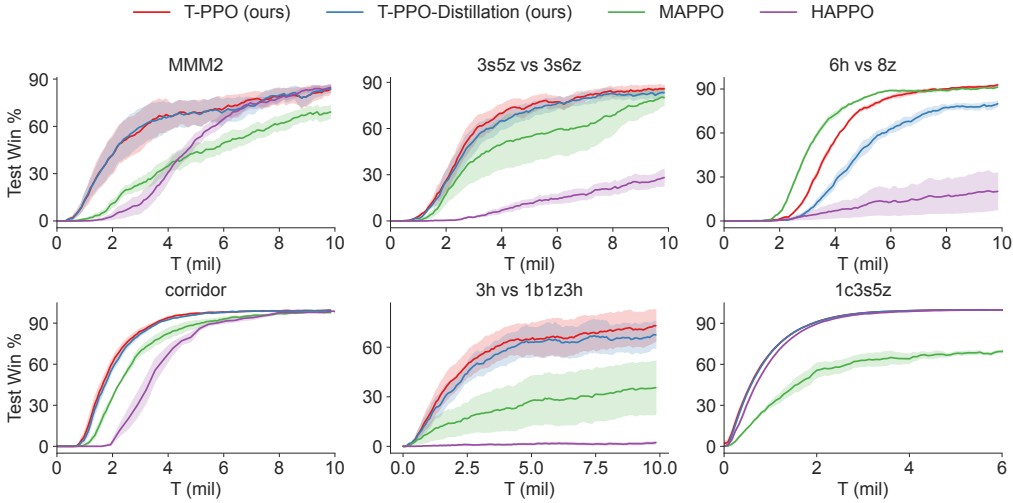

Figure 3: Learning curve of SMAC

Here we compare our approach with policy-based baselines on four super hard maps (MMM2, 3s5z_vs_3s6z, 6h_vs_8z, corridor), one easy map (1c3s5z), and one custom map (3h_vs_1b1z3h) based on the SMAC benchmark.

We illustrate the learning curve of StarCraft II in Figure 3. Our "single-agent" policy outperforms baselines on five out of six maps while performing similarly with MAPPO on 6h_vs_8z. Super hard maps are typically hard-exploration tasks. However, taking benefit of our approach's global optimality guarantee for MMDP, T-PPO can exploit better with the same exploration strategy as MAPPO (based on the entropy of learned policies). We will highlight a map 1c3s5z, where MAPPO con-

verges to a locally optimal point but our approach achieves global optimality with nearly 100% winning rate. This phenomenon once again demonstrates the advantage of our sequential transformation framework. HAPPO can also achieve a nearly 100% winning rate on `1c3s5z` but fails on other maps. We believe it is still because local optimality of Nash equilibrium learned by HAPPO. Meanwhile, compared with our approach and MAPPO, agents cannot share parameters in HAPPO, which significantly affects the training efficiency in complex tasks.

Table 2: Mean value and standard deviation of winning rate on the SMAC benchmark.

| Task | Difficulty | T-PPO | T-PPO-D | MAPPO | HAPPO | QPLEX |
|---|---|---|---|---|---|---|
| MMM2 | Super Hard | **81.6 (7.7)** | **82.5 (6.1)** | 68.5 (8.4) | **83.9 (1.5)** | 74.8 (11.6) |
| 3s5z vs 3s6z | Super Hard | **85.5 (5.2)** | 82.7 (6.2) | 78.3 (14.5) | 26.0 (4.3) | 75.8 (12.3) |
| 6h vs 8z | Super Hard | **91.8 (1.1)** | 78.6 (4.1) | **90.7 (0.7)** | 20.0 (13.3) | 20.7 (14.7) |
| corridor | Super Hard | **99.1 (0.3)** | **99.2 (0.1)** | 97.9 (0.4) | **99.2 (0.2)** | 30.3 (41.3) |
| 3h vs 1b1z3h | Local Optimal | **71.4 (20.7)** | 65.9 (17.2) | 35.0 (31.6) | 1.9 (1.4) | 30.7 (23.2) |
| 1c3s5z | Easy | **99.8 (0.0)** | **99.8 (0.1)** | 73.3 (11.4) | **99.9 (0.0)** | **98.7 (1.0)** |

We show the mean and standard deviation between the winning rates of the final policies trained on different random seeds in Table 2. T-PPO-Distillation (T-PPO-D, decentralized policy) outperforms T-PPO in two maps while outperforming MAPPO in five out of six maps, indicating our approach's competitiveness in decentralized execution. Only in `6h_vs_8z`, T-PPO-Distillation performs worse than T-PPO, demonstrating that correlation learned by T-PPO might be overly dependent on the sequential transformation even though we have added several normalization items. Although this phenomenon rarely occurs and depends on one special task, we will study this limitation in future work.

### 5.1.1 ADVANTAGE OF SEQUENTIAL TRANSFORMATION FRAMEWORK IN SMAC

In this section, we will further analyze the local optimality of SMAC. Compared with the matrix game, the StarCraft II tasks are more complex with high dimensional state space. To verify whether our approach can drive agents out of local optimal points as in matrix games, we create a new StraCraft II map named `3h_vs_1b1z3h`.

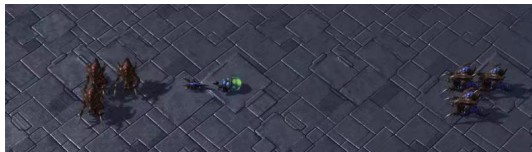

Figure 4: Illustration of `3h_vs_1b1z3h`.

In `3h_vs_1b1z3h`, we control three Hydralisks, while our opponent controls three Hydralisks, one low-damage Zergling, and one Baneling with high area damage. The explosion of the Baneling can only be avoided if all agents gather fire to it instead of the nearer Zergling. However, gathering fire to the Zergling could be a suboptimal equilibrium, where no agents tend to change its policy.

Previous research has demonstrated that super-hard maps in SMAC require more exploration (Wang et al., 2020; Li et al., 2021a). However, we find different experimental results on the maps that contain local optimal points, such as our own designed map `3h_vs_1b1z3h`. In PPO, exploration is guaranteed by the entropy term in its loss function. As shown in Figure 5, T-PPO and MAPPO's performance changes are divergent on different maps while tuning the related hyperparameter. On `6h_vs_8z`, increasing entropy weight will improve learning efficiency for both algorithms. It is in line with our expectations. On `3h_vs_1b1z3h`, increasing entropy weight will still improve the performance of T-PPO but will make the performance

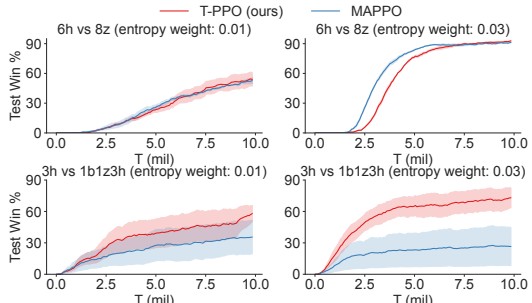

Figure 5: Different changes caused by adjusting the exploration coefficients (entropy weight) between T-PPO and MAPPO on `3h_vs_1b1z3h` and `6h_vs_8z`.

of MAPPO worse. We believe this phenomenon is related to the local optimality we discussed above. MAPPO has no motion to drive agents to escape local optimal points, which leads to low exploita-

tion efficiency. Taking advantage of handling global optimality as discussed in Section 4.1, our approach achieves better exploitation under the same exploration strategy.

## 5.2 GOOGLE RESEARCH FOOTBALL

In this section, we test our approach against policy-based baselines on another MARL benchmark named Google Research Football (GRF). In the environment setting, we use sparse rewards with both SCORING and CHECKPOINT for our approach and all baselines. For observations, we follow (Li et al., 2021a), using the simple 115-dimensional vector as the observation while removing the information irrelevant to the current scenario. Meanwhile, we introduce the relative position for each agent instead of absolute coordinates to achieve a more realistic environment.

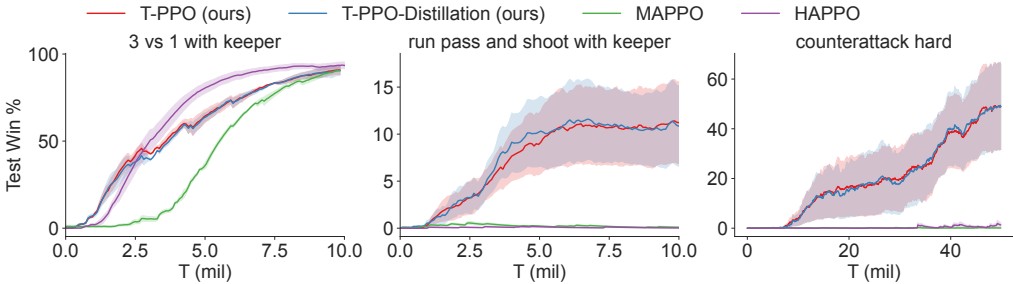

Figure 6: Learning curve of GRF.

As shown in Figure 6, our approach obviously outperforms baselines, achieving remarkable winning rate, while baselines almost learn nothing to win in `academy_pass_and_shoot_with_keeper` and `academy_counterattack_hard`. In GRF scenarios, agents must coordinate timing and positions to organize offense to seize fleeting opportunities. The cooperation between agents is difficult to coordinate because of the sparsity of agents' crucial movements. Our sequential transformation framework provides an optimal solution to MMDP by forcing agents to consider the information from previous ones, which promotes coordination among agents for achieving sophisticated cooperation. Compared with T-PPO, T-PPO-Distillation performs similarly, which ensures that our algorithm can be executed in a wide range of environments.

## 6 CONCLUSION

In this paper, we study state-of-the-art cooperative multi-agent reinforcement learning methods and observe that, even with full expressiveness, they may fail to converge to an optimal solution in simple matrix games. To analyze this phenomenon, we generalize these MARL methods with a general model and find that their factorized policy structure combined with the gradient descent optimization is one of the major causes of their suboptimality. To solve this issue, we propose a novel sequential transformation framework that allows employing off-the-shelf single-agent reinforcement learning methods to solve cooperative multi-agent tasks and retain their global optimality guarantee. Based on this framework, we develop T-PPO that extends single-agent PPO to multi-agent settings and significantly outperforms baselines on various benchmark tasks. It is an interesting future direction to extend efficient value-based SARL methods to multi-agent settings through our transformation framework.

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

# A    APPENDIX

## A.1    SUBOPTIMALITY OF EXISTING CTDE ALGORITHMS

For multi-agent actor-critic algorithms, we recall Theorem 3.1:

**Theorem 3.1.** *For multi-agent actor-critic algorithms, any Nash's Equilibrium of policies is a stationary point of actor loss. Moreover, there exists a family of single step-MMDP such that the actor loss function contains $\Omega(|\mathcal{A}|)$ different local minimums for deterministic policy and infinite local minimums for stochastic policy.*

*Proof.* Suppose $(\pi_{\theta_1}, \cdots, \pi_{\theta_n})$ is an NE, then by the definition of NE, we have $\forall i = 1, \cdots, n :$ $\forall \theta_i' : \mathcal{J}(\pi_{\theta_1}, \cdots, \pi_{\theta_n}) \geq \mathcal{J}(\pi_{\theta_1}, \cdots, \pi_{\theta_{i-1}}, \pi_{\theta_i'}, \pi_{\theta_{i+1}}, \cdots, \pi_{\theta_{-i}})$.

By denoting $C$ as the actor loss function, it's equivalent to $\forall i = 1, \cdots, n : \forall \theta_i' : C(\theta_1, \cdots, \theta_n) \leq C(\theta_1, \cdots, \theta_{i-1}, \theta_i', \theta_{i+1}, \cdots, \theta_{-i})$.

Suppose $\exists i : \frac{\partial C}{\partial \theta_i} \neq \mathbf{0}$. Let $l = (\underbrace{\mathbf{0}, \cdots, \mathbf{0}}_{i-1}, \frac{\partial C}{\partial \theta_i}, \underbrace{\mathbf{0}, \cdots, \mathbf{0}}_{n-i})$, we have

$$\lim_{\delta \to 0} \frac{C((\theta_1, \cdots, \theta_n) + \delta l) - C(\theta_1, \cdots, \theta_n)}{\delta \|l\|_2} = \|l\|_2 \neq 0$$

Choose a sufficient small $\delta$ will constitute a contradiction of the definition of NE. Thus we have $\forall i : \frac{\partial C}{\partial \theta_i} = \mathbf{0}$.

After that, by applying the derivation rule of compound function,

$$\frac{\partial C}{\partial \Theta} = \sum_{i=1}^{n} \frac{\partial C}{\partial \theta_i} \frac{\partial \theta_i}{\partial \Theta} = \mathbf{0}$$

we can address the situation where parameter-sharing is taken into consideration, which completes our proof of the first part of the theorem.

For the second part of the theorem, we construct a 2-agent matrix game here first. The payoff matrix $M$ of the matrix game is

$$M = \begin{pmatrix} |\mathcal{A}| & -K & \cdots & -K \\ -K & |\mathcal{A}| - 1 & \cdots & -K \\ \vdots & \vdots & \ddots & \vdots \\ -K & -K & \cdots & 1 \end{pmatrix}$$

where $K > 0$ is a positive constant.

In this case, any entry of the diagonal is a local minimum. Let $\boldsymbol{p}, \boldsymbol{q}$ be the $i$-th one-hot probability vector for some $i = 1, \cdots, |\mathcal{A}|$. The payoff of joint policy $(\boldsymbol{p}, \boldsymbol{q})$ is $\mathcal{J}(\boldsymbol{p}, \boldsymbol{q}) = \boldsymbol{p}^\top M \boldsymbol{q}$. Let's disturb the joint policy a little, such that $\boldsymbol{p}_i', \boldsymbol{q}_i' \in (1 - \epsilon, 1 - \epsilon/2)$. Then denoting $\Delta \boldsymbol{p} = (\boldsymbol{p} - \boldsymbol{p}'), \Delta \boldsymbol{q} = (\boldsymbol{q} - \boldsymbol{q}')$, the disturbed payoff is

$$\begin{aligned} \boldsymbol{p}'^\top M \boldsymbol{q}' &= (\boldsymbol{p} - \Delta p)^\top M (\boldsymbol{q} - \Delta q) \\ &\leq \mathcal{J}(\boldsymbol{p}, \boldsymbol{q}) - O(\epsilon) + O(\epsilon^2) \\ &< \mathcal{J}(\boldsymbol{p}, \boldsymbol{q}) \end{aligned}$$

for sufficiently small $\epsilon$. This means $(\boldsymbol{p}, \boldsymbol{q})$ is a local minimum. This case is easy to extent to general MMDPs.

$\square$

For value-decomposition algorithms, we recall Theorem 3.2. Before we prove the theorem, we prove a stronger version for a special case (QPLEX) first.

**Proposition A.1.** *Assuming the neural network as a universal approximator, for any MMDP with no cycle, there are infinite many local optima of the TD-loss function of QPLEX.*

*Proof.* We first expand the original formula of QPLEX (Wang et al., 2021b):

$$
\begin{aligned}
Q(s, \boldsymbol{a}) &= V_{tot}(s) + A_{tot}(s, \boldsymbol{a}) \\
&= \sum V_i(s) + \sum \lambda_i(s, \boldsymbol{a}) A_i(s, a_i) \\
&= \sum (w_i(s) V_i(s) + b_i(s)) + \sum \lambda_i(s, \boldsymbol{a}) A_i(s, a_i)
\end{aligned}
$$

where $w_i : \mathcal{S} \to \mathbb{R}, b_i : \mathcal{S} \to \mathbb{R}, Q_i : \mathcal{S} \times \mathcal{A}^n \to \mathbb{R}$ are neural networks, $V_i(s) = \max Q_i(s, \cdot), A_i(s, a_i) = Q_i(s, a_i) - V_i(s)$.

By assuming the neural network as a universal approximator and a little abuse of notations, the original formula in QPLEX can be rewritten as follows:

$$
Q(s, \boldsymbol{a}) = b(s) + \sum \lambda_i(s, \boldsymbol{a}) A_i(s, a_i)
$$

where $b, \lambda_i, Q_i$ are parameterized universal approximators, and $A_i(s, a_i) = Q_i(s, a_i) - \max Q_i(s, \cdot)$ is the individual advantage function.

We explicitly specify the parameters used by each approximator here: $b(s; \psi)$, $\lambda_i(s, \boldsymbol{a}; \phi_i)$, $Q_i(s, a_i; \theta_i)$. Denoting $\Theta = (\theta_1, \cdots, \theta_n, \phi_1, \cdots, \phi_n, \psi)$ as all parameters being used, the TD-loss function is

$$
\mathcal{L}_{TD}(\Theta) = \frac{1}{2} \mathbb{E}_{(s,a) \sim \mathcal{D}} \left[ Q(s, \boldsymbol{a}; \Theta) - (\mathcal{T} Q)(s, \boldsymbol{a}) \right]^2
$$

for some distribution $\mathcal{D}$ fully supported on $\mathcal{S} \times \mathcal{A}^n$, where $\mathcal{T}$ is the Bellman operator.

The distribution $\mathcal{D}$ does not matters that much here as long as it is fully supported. We now further assume $\mathcal{D}$ to be a uniform distribution. For $\mathcal{D}$ that is not uniform, the proof is essentially similar. We leave it to the reader.

Fix any non-degenerated $\theta_1, \cdots, \theta_n$, which means $\forall i, s$, the greedy action of $Q_i(s, \cdot)$ is unique. Then a sufficiently small neighborhood of $\Theta$ won't change the greedy joint action due to the continuity of the function [1]. Therefore, we can fix $\boldsymbol{a}^*(s) = \arg\max Q(s, \cdot)$ for each state $s$ ($\boldsymbol{a}^*(s)$ is abbreviated as $\boldsymbol{a}^*$ when there is no ambiguity).

Denote $T(s, \boldsymbol{a}) = (\mathcal{T} Q)(s, \boldsymbol{a})$ as the current target in TD-loss.

We first consider the case when the MMDP is an one-step game (i.e. $\gamma = 0$). Then $T$ is a constant tensor independent to current $Q$ in this case. The TD-learning becomes a supervised learning task.

We try to optimize the TD-loss by minimizing $\sum_{\boldsymbol{a}} [Q(s, \boldsymbol{a}; \Theta) - T(s, \boldsymbol{a})]^2$ for each $s$.

Then for any state $s$, denote $L = \{\boldsymbol{a} \neq \boldsymbol{a}^* : T(s, \boldsymbol{a}) \leq T(s, \boldsymbol{a}^*)\}, G = \{\boldsymbol{a} : T(s, \boldsymbol{a}) > T(s, \boldsymbol{a}^*)\} \cup \{\boldsymbol{a}^*\}$. We have

$$
\begin{aligned}
&\sum_{\boldsymbol{a}} [Q(s, \boldsymbol{a}; \Theta) - T(s, \boldsymbol{a})]^2 \\
&= \sum_{\boldsymbol{a} \in L} [Q(s, \boldsymbol{a}; \Theta) - T(s, \boldsymbol{a})]^2 + \sum_{\boldsymbol{a} \in G} [Q(s, \boldsymbol{a}; \Theta) - T(s, \boldsymbol{a})]^2
\end{aligned}
$$

---

[1] Note that the continuity is naturally assumed, since we need to take the gradient of the loss function w.r.t. $\Theta$

Keeping $\theta_1, \cdots, \theta_n$ fixed, find $\psi, \phi_1, \cdots, \phi_n$ such that

$$b(s; \psi) = \frac{1}{|G|} \sum_{\boldsymbol{a}} \max(T(s, \boldsymbol{a}) - T(s, a^*), 0) + T(s, \boldsymbol{a}^*)$$

$$\forall i : \lambda(s, \boldsymbol{a}; \phi_i) = \min(T(s, \boldsymbol{a}) - T(s, \boldsymbol{a}^*), 0) / \sum_i A_i(s, a_i; \theta_i)$$

Plug these formula into the definition of mixing network, we have

$$\forall \boldsymbol{a} \in L : Q(s, \boldsymbol{a}; \Theta) = T(s, \boldsymbol{a})$$

$$\forall \boldsymbol{a} \in G : Q(s, \boldsymbol{a}; \Theta) = \frac{1}{|G|} \sum_{\boldsymbol{a}' \in G} T(s, \boldsymbol{a}')$$

It's easy to verify that

$$\sum_{\boldsymbol{a} \in L} [Q(s, \boldsymbol{a}; \Theta) - T(s, \boldsymbol{a})]^2 = 0$$

and

$$\sum_{\boldsymbol{a} \in G} [Q(s, \boldsymbol{a}; \Theta) - T(s, \boldsymbol{a})]^2 = \min_{v \in \mathbb{R}} \sum_{\boldsymbol{a} \in G} [v - T(s, \boldsymbol{a})]^2$$

These further show that

$$\mathcal{L}_{TD}(\Theta) = \min_{\mathcal{Q} \in \mathbb{R}^{|\mathcal{S}| \times |\mathcal{A}|^n} : \forall s : \boldsymbol{a}^*(s) \in \arg\max \mathcal{Q}(s, \cdot)} [\mathcal{Q}(s, \boldsymbol{a}) - T(s, \boldsymbol{a})]^2$$

which means $\Theta$ is a global minimum conditioning on that the greedy joint policy stays unchanged. Recall that there is a small neighborhood of $\Theta$ such that the greedy joint policy stays unchanged, this finishes the proof of local optimality of $\Theta$ in $\mathcal{L}_{TD}$.

This argument can be easily extended to acyclic MMDPs. We just need to calculate $\psi$ by the reversal of topological order, then all formula above will be well-defined.

$\square$

Now we are able to prove Theorem 3.2

**Theorem 3.2.** *There exists a family of MMDP, such that for any value-decomposition algorithm with a complete Q-function class satisfying the IGM condition (Eq. (3)), the TD-loss function contains* $\Omega\left(|A|^{|\mathcal{S}|}\right)$ *different local optima.*

*Proof.* Recall Eq. (2), the mixing network of any value-decomposition algorithm on MMDP is in the following form

$$Q(s, \boldsymbol{a}; \Theta) = f_{\text{mix}}(Q_1(s, \cdot), \cdots, Q_n(s, \cdot), s, \boldsymbol{a}; \Theta)$$

Following the proof of Proposition A.1, it's essential to to find a series of $\Theta^{(k)}$ such that (1) $\Theta^{(k)}$ is globally optimal (w.r.t the TD-loss function) conditioning on that the greedy joint policy unchanged; (2) the greedy joint policy is unchanged in a small neighborhood of $\Theta^{(k)}$.

Now we construct the MMDP as follows: a. let $\gamma = 0$; b. let $r(s, \boldsymbol{a}) = \begin{cases} i, & a_1 = a_2 = \cdots = a_n = i \\ 0, & \text{otherwise} \end{cases}$ for all $s \in \mathcal{S}$. Since $\gamma = 0$, the transition probability does not really matter.

Let $\Pi = \{\boldsymbol{\pi} : \mathcal{S} \to \mathcal{A} : \pi_1 = \pi_2 = \cdots = \pi_n\}$ be the set of policies with non-zero value on each state. Note that $|\Pi| = |\mathcal{A}|^{|\mathcal{S}|}$. If we can show that there is a $\Theta^{(\pi)}$ for each $\pi \in \Pi$ satisfying (1) and (2), then we finish our proof.

Fix any $\pi \in \Pi$, we construct $\Theta^{(\pi)}$ as follows.

First, let $f_m(s, \boldsymbol{a}) = \begin{cases} \frac{a_1 + |\mathcal{A}|}{2}, & \boldsymbol{a} = \pi(s) \\ \frac{\pi(s)_1 + |\mathcal{A}|}{2} - \frac{1}{m}, & a_1 = a_2 = \cdots = a_n > \pi(s)_1 \\ a_1, & a_1 = a_2 = \cdots = a_n < \pi(s)_1 \\ 0, & \text{otherwise} \end{cases}$ for $m \in \mathbb{N}$.

By the completeness of $Q$-function class, we are able to find some $\Theta_m$, such that $Q(s, a; \Theta_m) = f_m(s, \boldsymbol{a})$. By the uniqueness of greedy policy of $f_m$, we have $\forall a_i \in \mathcal{A} : Q_i(s, \pi(s); \Theta_m) > Q_i(s, a_i; \Theta_m)$ for $i = 1, \cdots, n$.

By Bolzano Weierstrass Theorem[2], we can find a convergent subsequence $\{\Theta_{m_k}\}_{k=1}^{\infty}$. Take the limit, we have $\Theta_{m_k} \to \Theta^{(\pi)}$. By the continuity[3], we have $Q(s, a; \Theta^{(\pi)}) = \begin{cases} \frac{\pi(s)_1 + |\mathcal{A}|}{2}, & a_1 = a_2 = \cdots = a_n \geq \pi(s)_1 \\ a_1, & a_1 = a_2 = \cdots = a_n < \pi(s)_1 \\ 0, & \text{otherwise} \end{cases}$, and $\forall a_i \in \mathcal{A} : Q_i(s, \pi(s); \Theta^{(\pi)}) \geq Q_i(s, a_i; \Theta^{(\pi)})$ for $i = 1, \cdots, n$.

(1) is obvious in this case. We can prove it by following the argument in the proof in Proposition A.1.

To prove (2), we need to prove the strict inequivalence in $\forall a_i \in \mathcal{A} : Q_i(s, \pi(s); \Theta^{(\pi)}) \geq Q_i(s, a_i; \Theta^{(\pi)})$. We prove it by contradiction.

Suppose that there are some $s, i$, and $a_i$, such that $Q_i(s, \pi(s); \Theta^{(\pi)}) = Q_i(s, a_i; \Theta^{(\pi)})$. Then we have $Q(s, \pi(s)_1, \cdots, a_i, \cdots, \pi(s)_n; \Theta^{(\pi)}) = \max Q(s, \cdot; \Theta^{(\pi)}) = \frac{\pi(s)_1 + |\mathcal{A}|}{2}$ by the IGM assumption, which contradicts to $Q(s, \pi(s)_1, \cdots, a_i, \cdots, \pi(s)_n; \Theta^{(\pi)}) = 0$. $\square$

**Experimental Results for QPLEX** In empirical design of the algorithm, we notice that QPLEX has used some engineering tricks like "stop gradient" to modify the gradient of non-optimal points helping the algorithm to jump out of some local optimums. But these tricks are lack of theoretical guarantee, we can still construct cases that QPLEX is not able to reach global optimum, such as the following Matrix Game (Table3).

| -20 | 10 |
|-----|----|
| 10  | 9  |

Table 3: Matrix Game 2, $m = 2$

This Matrix Game has two global optimums $(0, 1)$ and $(1, 0)$, and a suboptimal solution $(1, 1)$ with high reward. QPLEX will likely to initialize to the suboptimal solution $(1, 1)$, and after that, it get confused since the manually modified gradient doesn't tell it the right direction. The learnt joint $Q$ vibrates around the following matrix:

$$\begin{pmatrix} -20 & 29/3 - \epsilon \\ 29/3 - \epsilon & 29/3 \end{pmatrix}$$

which can be proved to be a local optimum of $\mathcal{L}_{TD}$ according to the proof of Proposition A.1.

---

[2]BW Theorem claims that any bounded sequence in $\mathbb{R}^k$ has a convergent subsequence. It's sufficient to assume the boundedness for the purpose of understanding the main idea of the proof. For the unbounded cases, this theorem can be extended a little bit to deal with unbounded sequence by defining the convergence of $x_n / |x_n|$ as an extended convergence. This is a standard technique in mathematical analysis, we omit it here for conciseness.

[3]Note that the continuity is naturally assumed, since we need to take the gradient of the loss function w.r.t. $\Theta$

**QTRAN** $\mathcal{L}_{TD}$ in QTRAN is discontinuous, for everywhere the police switches may constitute a jump discontinuity, which may be harmful for gradient descent methods, since gradient descent methods assume the loss function to be differentiable. Unfortunately, we are not able to give any theoretical analysis about QTRAN, either prove or disprove its optimality. We have only empirical results shown in Section 5 to prove its potential suboptimality.

## A.2 SEQUENTIAL TRANSFORMATION

We present the formal definition of sequential transformation here for completeness.

**Definition A.1** (Sequential Transformation $\Gamma$). *Given an MMDP $\mathcal{M} = (\mathcal{S}, \mathcal{A}, P, r, \gamma, s_0, N)$, its sequential transformation is an MDP $\Gamma(\mathcal{M}) = \left(\tilde{\mathcal{S}}, \mathcal{A}, \tilde{P}, \tilde{r}, \tilde{\gamma}, s_0\right)$, where $\tilde{\mathcal{S}} = \bigcup_{i=0}^{N-1} \mathcal{S} \times \mathcal{A}^i$ is the state space, $\mathcal{A}$ is the same action space as the original MMDP $M$, $\tilde{P}$ is the transformed transition function, where $\forall k < N, \forall \tilde{s} = (s, a_1, \cdots, a_{k-1}) \in \tilde{\mathcal{S}}, \forall a_k \in \mathcal{A}$, we have $\tilde{P}((s, a_1, \cdots, a_k)|\tilde{s}, a_k) = 1$, and $\forall \tilde{s} = (s, a_1, \cdots, a_{N-1}) \in \tilde{\mathcal{S}}, \forall s' \in \mathcal{S}, \forall a_N \in \mathcal{A}$, we have $\tilde{P}(s'|\tilde{s}, a_N) = P(s'|s, (a_1, \cdots, a_N))$, $\tilde{r}$ is the transformed reward function, where $\forall k < N, \forall \tilde{s} = (s, a_1, \cdots, a_{k-1}) \in \tilde{\mathcal{S}}, \forall a_k \in \mathcal{A}$, we have $\tilde{r}(\tilde{s}, a_k) = 0$, and $\forall \tilde{s} = (s, a_1, \cdots, a_{N-1}) \in \tilde{\mathcal{S}}, \forall a_N \in \mathcal{A}$, we have $\tilde{r}(\tilde{s}, a_N) = r(s, (a_1, \cdots, a_N))$, $\tilde{\gamma} = \gamma^{1/N}$ is a transformed discount factor, and $s_0$ is the initial state.*

### A.2.1 PSEUDOCODE OF THE FRAMEWORK WITH SEQUENTIAL TRANSFORMATION

Here we present pseudo-code of the sequential framework (Algorithm 1).

---
**Algorithm 1** The Sequential Framework
---
1: **Input:** An SARL algorithm $\mathfrak{A}$, an oracle $O^{\mathcal{M}}$ for interaction with an MMDP $\mathcal{M}$.
2: **while** Simulating $\mathfrak{A}$ **do**
3:     **if** $\mathfrak{A}$ asks for the initialization of environment **then**
4:         Initialize $t = 0$, $a_i = 0$ for $i = 1, \cdots, N$.
5:         Call $O^{\mathcal{M}}$ for the initialization of $\mathcal{M}$ and obtain $s_0$
6:         Return $s_0$ to $\mathfrak{A}$.
7:     **else if** $\mathfrak{A}$ asks for an interaction with the environment by providing an action $a$ **then**
8:         $a_{t \bmod N+1} \leftarrow a$
9:         **if** $t \bmod N = N - 1$ **then**
10:            Call $O^{\mathcal{M}}$ for the interaction by providing action $(a_1, \cdots, a_N)$ to obtain reward $r$ and next state $s'$.
11:            $s \leftarrow s'$
12:            Return $r$ and $s'$ to $\mathfrak{A}$.
13:         **else**
14:            Return 0 and $(s, a_1, \cdots, a_{t \bmod N+1})$ to $\mathfrak{A}$.
15:         **end if**
16:         $t = t + 1$
17:     **else if** $\mathfrak{A}$ returns an policy $\pi$ **then**
18:         Convert $\pi$ to the joint policy $\pi_{\text{jt}}$ on $\mathcal{M}$.
19:         **Break**
20:     **end if**
21: **end while**
22: **Return** $\pi_{\text{jt}}$.

---

### A.2.2 THE EQUIVALENCE BETWEEN $\mathcal{M}$ AND $\Gamma M$ AS WELL AS THE CONVERSION OF POLICIES

Here we state the theorem of equivalence between $\mathcal{M}$ and $\Gamma\mathcal{M}$ in perspective of policy value.

**Theorem A.1.** *For any deterministic policy $\pi$ on $\Gamma(\mathcal{M})$, there is a decentralized policy $\pi_{jt} = (\pi_1, \cdots, \pi_N)$ on $\mathcal{M}$ such that $\mathcal{J}_{\mathcal{M}}(\pi_{jt}) = \gamma^{(1-n)/n} \mathcal{J}_{\Gamma(\mathcal{M})}(\pi)$, where $\pi_1(s) = \pi(s)$, $\pi_k(s) = \pi((s, \pi_1(s), \cdots, \pi_{k-1}(s)))$ for all $k > 1$.*

*For any stochastic policy $\eta$ on $\Gamma(\mathcal{M})$, there is a communicated policy $\eta_{\mathrm{jt}} = (\eta_1, \cdots, \eta_N)$ on $\mathcal{M}$ such that $\mathcal{J}_{\mathcal{M}}(\eta_{\mathrm{jt}}) = \gamma^{(1-n)/n}\mathcal{J}_{\Gamma(\mathcal{M})}(\eta)$, where $\eta_1(a_1|s) = \eta(a_1|s)$, $\eta_k(a_k|s, a_1, \cdots, a_{k-1}) = \eta(a_k|(s, a_1, \cdots, a_{k-1}))$ for all $k > 1$, where $a_1, \cdots, a_{k-1}$ are actions selected by agents $1, \cdots, k-1$.*

*And conversely, for any policy $\pi_{\mathrm{jt}}$ on $\mathcal{M}$, there is a policy $\pi$ on $\Gamma\mathcal{M}$ such that $\mathcal{J}_{\mathcal{M}}(\pi_{\mathrm{jt}}) = \gamma^{(1-n)/n}\mathcal{J}_{\Gamma(\mathcal{M})}(\pi)$.*

*Proof.* For deterministic policy:

$$
\begin{aligned}
\mathcal{J}_{\mathcal{M}}(\pi_{\mathrm{jt}}) &= \mathbb{E}\left[\sum_{t=0}^{\infty}\gamma^t r(s_t, \pi_{\mathrm{jt}})\middle| s_{t+1} \sim P(s_t, \pi_{\mathrm{jt}})\right] \\
&= \mathbb{E}\left[\sum_{t=0}^{\infty}\tilde{\gamma}^{nt}\tilde{r}((s_t, \pi_1(s_t), \cdots, \pi_{n-1}(s_t)), \pi)\middle| s_{t+1} \sim P(s_t, \pi_{\mathrm{jt}})\right] \\
&= \mathbb{E}\left[\sum_{t=0}^{\infty}\sum_{k=0}^{n-1}\tilde{\gamma}^{nt+k-n+1}\tilde{r}((s_t, \pi_1(s_t), \cdots, \pi_{k-1}(s_t)), \pi)\middle| s_{t+1} \sim P(s_t, \pi_{\mathrm{jt}})\right] \\
&= \mathbb{E}\left[\tilde{\gamma}^{1-n}\sum_{t'=0}^{\infty}\tilde{\gamma}^{t'}\tilde{r}(s_{t'}, \pi)\middle| s_{t'+1} \sim \tilde{P}(s_{t'}, \pi)\right] \\
&= \gamma^{\frac{1-n}{n}}J_{\Gamma(\mathcal{M})}(\pi)
\end{aligned}
$$

For stochastic policy the proof is similar,

$$
\begin{aligned}
\mathcal{J}_{\mathcal{M}}(\eta_{\mathrm{jt}}) &= \mathbb{E}\left[\sum_{t=0}^{\infty}\gamma^t r(s_t, \eta_{\mathrm{jt}})\middle| s_{t+1} \sim P(s_t, \eta_{\mathrm{jt}})\right] \\
&= \mathbb{E}\left[\sum_{t=0}^{\infty}\tilde{\gamma}^{nt}\tilde{r}\left(\left(s_t, a_{<n}^{(t)}\right), \eta\right)\middle| s_{t+1} \sim P(s_t, \eta_{\mathrm{jt}}), a_l^{(t)} \sim \eta\left(\cdot\middle|\left(s_t, a_{<l}^{(t)}\right)\right)\right] \\
&= \mathbb{E}\left[\sum_{t=0}^{\infty}\sum_{k=0}^{n-1}\tilde{\gamma}^{nt+k-n+1}\tilde{r}\left(\left(s_t, a_{<k}^{(t)}\right), \eta\right)\middle| s_{t+1} \sim P(s_t, \eta_{\mathrm{jt}}), a_l^{(t)} \sim \eta\left(\cdot\middle|\left(s_t, a_{<l}^{(t)}\right)\right)\right] \\
&= \mathbb{E}\left[\tilde{\gamma}^{1-n}\sum_{t'=0}^{\infty}\tilde{\gamma}^{t'}\tilde{r}(s_{t'}, \eta)\middle| s_{t'+1} \sim \tilde{P}(s_{t'}, \eta)\right] \\
&= \gamma^{\frac{1-n}{n}}J_{\Gamma(\mathcal{M})}(\eta)
\end{aligned}
$$

For the converse part of the theorem, suppose $\pi_{\mathrm{jt}} = (\pi_1, \cdots, \pi_n)$, it's sufficient to let $\pi((s, a_1, \cdots, a_{k-1})) = \pi_k(s)$. The calculation of its value is similar to the above, we omit it here.

$\square$

## A.3 Optimality of TPPO

The proof of Theorem 4.1 and Proposition 4.1 are directly followed by Theorem A.1 and Theorem 4.9 in (Liu et al., 2019), which is omitted here.

## A.4 The Complexity of the Transformed Model

By sequential transform, we are able to convert any MMDP to an MDP and run SARL algorithms on the MDP to solve the MMDP. One natural question is, will such framework bring additional hardness of the task?

First of all, this framework obviously doesn't increase the minimax sample complexity of the task, since the MMDP $\mathcal{M}$ and the MDP $\Gamma\left(\mathcal{M}\right)$ can be transformed to each other with merely negligible additional cost in time and space (see Appendix A.6). Nevertheless, for a concrete algorithm $A$ (e.g. Q-learning), the sample complexity is not necessary to be the same after such transformation.

Let's take Q-learning as an example. We first investigate the size of state-action space before and after the transformation. It easy to see that the size of the state-action space of $\mathcal{M}$ is $|\mathcal{S}||\mathcal{A}|^N$, and that of its $\Gamma\left(\mathcal{M}\right)$ is $|\mathcal{A}|\sum_{i=0}^{N-1}|\mathcal{S}||\mathcal{A}|^i = |\mathcal{S}||\mathcal{A}|\frac{1-|\mathcal{A}|^N}{1-|\mathcal{A}|} \leq 2|\mathcal{S}||\mathcal{A}|^N$. This implies that the sequential transform does not increase the complexity in the state-action space.

However, if we take a closer look here of the sample complexity, we will find that the exact sample complexity bound of Q-learning is $O^*\left(\frac{|\mathcal{S}||\mathcal{A}|}{(1-\gamma)^4\epsilon^2}\right)$ ((Li et al., 2021b)), which depends on not only the size of state-action space, but also on the magnitude of $\frac{1}{1-\gamma}$. This implies that the sample complexity may increase for certain algorithms since $\Gamma\left(\mathcal{M}\right)$ has a longer horizon. Despite this unpleasant result, for Q-learning, still, this analysis leave out the structure of $\Gamma\left(\mathcal{M}\right)$: for every $n$ steps in $\Gamma\left(\mathcal{M}\right)$ there are $n-1$ deterministic transitions with reward 0. So fortunately, if we modify the original Q-learning a little bit, it will attain the same sample complexity as before (See Appendix A.5).

## A.5 AN EXTENSION OF Q-LEARNING

Here we introduce a variant of Q-learning dealing with deterministic transitions (Algorithm 2) to demonstrate the claim at the end of Appendix A.4. If we adopt this algorithm in our sequential transformation framework, it will have the same sample complexity as the original Q-learning on the original MMDP $\mathcal{M}$.

We denote $D : \mathcal{S} \times \mathcal{A} \rightarrow \{0,1\}$ as an oracle telling whether this state-action pair would result in a deterministic transition.

---

**Algorithm 2** QLDT (Q-learning for MDPs with deterministic transitions)

---

    **Access to:** An oracle $D : \mathcal{S} \times \mathcal{A} \rightarrow \{0,1\}$.
    Initialize $Q$
    **for** $t \leftarrow 0$ **to** $T$ **do**
      $s \leftarrow s_0$
      Initialize stack $v$.
      **for** each step of epoch $t$ **do**
        $a \leftarrow \text{SelectAction}(\pi, s)$ $\{\epsilon\text{-greedy}\}$
        $r \leftarrow \text{Reward}(s, a)$
        $s' \leftarrow \text{NextState}(s, a)$
        **if** $D(s, a)$ **then**
          Push $(s, a, r, s')$ into $v$ {deterministic transition}
          $Q(\tilde{s}, \tilde{a}) \leftarrow \tilde{r} + \gamma \max Q(\tilde{s}', \cdot)$
        **else**
          $Q(s, a) \leftarrow Q(s, a) + \alpha[r + \gamma \max Q(s', \cdot) - Q(s, a)]$
          **while** $v$ is not empty **do**
            Pop $v$ to get $(\tilde{s}, \tilde{a}, \tilde{r}, \tilde{s}')$
            $Q(\tilde{s}, \tilde{a}) \leftarrow \tilde{r} + \gamma \max Q(\tilde{s}', \cdot)$
          **end while**
        **end if**
        $s \leftarrow s'$
      **end for**
    **end for**

---

We have Proposition A.2.

**Proposition A.2.** *T-QLDT has the same sample complexity as original Q-learning on $\mathcal{M}$.*

*Proof.* One can view T-QLDT as the original Q-learning maintaining a max heap, where $Q((s, a_1, \cdots, a_{k-1}), a_k) = \max Q((s, a_1, \cdots, a_k), \cdot)$. In this way, T-QLDT has exactly the

same behaviour as the original Q-learning and does not change the sample complexity as a consequence. □

### A.6 THE MINIMAX SAMPLE COMPLEXITY OF THE SEQUENTIAL FRAMEWORK

Here we explain a bit more of the claim of the minimax sample complexity in Appendix A.4.

The minimax sample complexity here is the sample complexity of the "best" algorithm over the "hardest" task. For any multi-agent algorithm $A$, we can always find a single-agent algorithm $B$, such that $A = $ T-$B$. It is because that for any MDP $\widetilde{\mathcal{M}}$, we can always compress $n$ steps on $\widetilde{\mathcal{M}}$ into one step, and then use the corresponding multi-agent algorithm $A$ to solve it as a multi-agent problem. In this way, T-$B$ is exactly $A$, and thus not increase the minimax sample complexity. One should keep in mind that every $n$ samples on MDP correspond to exactly one sample on MMDP. Particularly, the number of bits we need to record every $n$ samples on MDP are exactly what we need to record one sample on MMDP.

## B EXPERIMENTAL DETAILS

In this section, we provide more experimental results supplementary to those presented in Section 5. We also discuss the details of the experimental settings of both our matrix game and the StarCraft II micromanagement (SMAC) benchmark.

### B.1 MULTI-TASK MATRIX GAME

In section 3, we design a multi-task matrix game to demonstrate the global optimality of our sequential transformation framework. In this section, we will first show the details of this environment and provide more evidence of our algorithms' advantage on MMDP.

#### B.1.1 DETAILS OF MULTI-TASK MATRIX GAME

In multi-task matrix game, the return of the optimal strategy corresponding to each matrix is 10, which means the sum rewards of the global optimal strategy is 100. Two agents are initialized to one matrix uniformly at random, and the ID of a current matrix is observable to both of them. They need to cooperate to select the entry with the maximum reward for the current matrix, after that, the game ends. Each matrix contains $5 \times 5 = 25$ entries, which means $\mathcal{A} = \{0, 1, 2, 3, 4\}$ for each agent.

All 10 payoff matrices are listed in Table 4. The optimal strategies' payoff of all matrices is 10. Matrices $1 - 5$ are hand-crafted in order to create some hard NEs. Matrices $6 - 10$ are drawn uniformly at random. It's worth noting that a random $5 \times 5$ matrix has $25/9 \approx 2.77$ different NEs in expectation. And in our opinion, the existence of suboptimal NEs is the main reason why existing algorithms fail.

#### B.1.2 VISUALIZATION OF LEARNED JOINT STRATEGIES

Table 5: Payoff matrix

| $a_2$ \ $a_1$ | $\mathcal{A}^{(1)}$ | $\mathcal{A}^{(2)}$ | $\mathcal{A}^{(3)}$ |
|---|---|---|---|
| $\mathcal{A}^{(1)}$ | 8 | -12 | -12 |
| $\mathcal{A}^{(2)}$ | -12 | 0 | 0 |
| $\mathcal{A}^{(3)}$ | -12 | 0 | 0 |

Table 6: $\pi_{tot}$ of TPPO

| $a_2$ \ $a_1$ | $\mathcal{A}^{(1)}$ | $\mathcal{A}^{(2)}$ | $\mathcal{A}^{(3)}$ |
|---|---|---|---|
| $\mathcal{A}^{(1)}$ | 1 | 0 | 0 |
| $\mathcal{A}^{(2)}$ | 0 | 0 | 0 |
| $\mathcal{A}^{(3)}$ | 0 | 0 | 0 |

Table 7: $\pi_{tot}$ of MAPPO

| $a_2$ \ $a_1$ | $\mathcal{A}^{(1)}$ | $\mathcal{A}^{(2)}$ | $\mathcal{A}^{(3)}$ |
|---|---|---|---|
| $\mathcal{A}^{(1)}$ | 0 | 0 | 0 |
| $\mathcal{A}^{(2)}$ | 0 | 0.2 | 0.2 |
| $\mathcal{A}^{(3)}$ | 0 | 0.3 | 0.3 |

To further illustrate our approach's ability to attach the global optimal point, we use the matrix game in (Wang et al., 2021b; Ma et al., 2021) as a toy example (shown in Table 5). Here we compare our approach T-PPO with MAPPO. Joint policies learned by T-PPO and MAPPO are shown in Table 6 and Table 7. MAPPO falls into local optimal points due to Theorem 3.1, while T-PPO obtains the optimal strategy. Taking this advantage, our approach dominates in our multi-task matrix game.

Table 4: Multi-task Matrix Game

Matrix 1

| $a_2$ \ $a_1$ | $\mathcal{A}^{(1)}$ | $\mathcal{A}^{(2)}$ | $\mathcal{A}^{(3)}$ | $\mathcal{A}^{(4)}$ | $\mathcal{A}^{(5)}$ |
|---|---|---|---|---|---|
| $\mathcal{A}^{(1)}$ | **10** | -10 | -10 | -10 | -10 |
| $\mathcal{A}^{(2)}$ | -10 | 9 | 0 | 0 | 0 |
| $\mathcal{A}^{(3)}$ | -10 | 0 | 9 | 0 | 0 |
| $\mathcal{A}^{(4)}$ | -10 | 0 | 0 | 9 | 0 |
| $\mathcal{A}^{(5)}$ | -10 | 0 | 0 | 0 | 9 |

Matrix 2

| $a_2$ \ $a_1$ | $\mathcal{A}^{(1)}$ | $\mathcal{A}^{(2)}$ | $\mathcal{A}^{(3)}$ | $\mathcal{A}^{(4)}$ | $\mathcal{A}^{(5)}$ |
|---|---|---|---|---|---|
| $\mathcal{A}^{(1)}$ | **10** | -10 | **10** | -10 | **10** |
| $\mathcal{A}^{(2)}$ | -10 | **10** | -10 | **10** | -10 |
| $\mathcal{A}^{(3)}$ | **10** | -10 | **10** | -10 | **10** |
| $\mathcal{A}^{(4)}$ | -10 | **10** | -10 | **10** | -10 |
| $\mathcal{A}^{(5)}$ | **10** | -10 | **10** | -10 | **10** |

Matrix 3

| $a_2$ \ $a_1$ | $\mathcal{A}^{(1)}$ | $\mathcal{A}^{(2)}$ | $\mathcal{A}^{(3)}$ | $\mathcal{A}^{(4)}$ | $\mathcal{A}^{(5)}$ |
|---|---|---|---|---|---|
| $\mathcal{A}^{(1)}$ | -20 | -20 | -20 | -20 | **10** |
| $\mathcal{A}^{(2)}$ | -20 | -20 | -20 | **10** | 9 |
| $\mathcal{A}^{(3)}$ | -20 | -20 | **10** | 9 | 9 |
| $\mathcal{A}^{(4)}$ | -20 | **10** | 9 | 9 | 9 |
| $\mathcal{A}^{(5)}$ | **10** | 9 | 9 | 9 | 9 |

Matrix 4

| $a_2$ \ $a_1$ | $\mathcal{A}^{(1)}$ | $\mathcal{A}^{(2)}$ | $\mathcal{A}^{(3)}$ | $\mathcal{A}^{(4)}$ | $\mathcal{A}^{(5)}$ |
|---|---|---|---|---|---|
| $\mathcal{A}^{(1)}$ | -20 | -20 | -20 | -20 | **10** |
| $\mathcal{A}^{(2)}$ | -20 | -20 | -20 | **10** | 9 |
| $\mathcal{A}^{(3)}$ | -20 | -20 | **10** | 9 | 8 |
| $\mathcal{A}^{(4)}$ | -20 | **10** | 9 | 8 | 7 |
| $\mathcal{A}^{(5)}$ | **10** | 9 | 8 | 7 | 6 |

Matrix 5

| $a_2$ \ $a_1$ | $\mathcal{A}^{(1)}$ | $\mathcal{A}^{(2)}$ | $\mathcal{A}^{(3)}$ | $\mathcal{A}^{(4)}$ | $\mathcal{A}^{(5)}$ |
|---|---|---|---|---|---|
| $\mathcal{A}^{(1)}$ | -20 | -15 | -10 | -5 | 6 |
| $\mathcal{A}^{(2)}$ | -20 | -15 | -10 | 7 | 5 |
| $\mathcal{A}^{(3)}$ | -20 | -15 | 8 | 6 | 4 |
| $\mathcal{A}^{(4)}$ | -20 | 9 | 7 | 5 | 3 |
| $\mathcal{A}^{(5)}$ | **10** | 8 | 6 | 4 | 2 |

Matrix 6

| $a_2$ \ $a_1$ | $\mathcal{A}^{(1)}$ | $\mathcal{A}^{(2)}$ | $\mathcal{A}^{(3)}$ | $\mathcal{A}^{(4)}$ | $\mathcal{A}^{(5)}$ |
|---|---|---|---|---|---|
| $\mathcal{A}^{(1)}$ | 0.8 | -16.0 | -5.0 | -10.9 | -3.7 |
| $\mathcal{A}^{(2)}$ | -9.2 | -4.2 | 7.3 | 9.6 | -3.0 |
| $\mathcal{A}^{(3)}$ | -20.0 | -18.1 | 0.2 | -4.3 | 9.0 |
| $\mathcal{A}^{(4)}$ | -14.9 | -2.0 | -17.7 | -17.6 | -0.8 |
| $\mathcal{A}^{(5)}$ | 3.8 | **10** | 7.5 | 9.2 | -10.7 |

Matrix 7

| $a_2$ \ $a_1$ | $\mathcal{A}^{(1)}$ | $\mathcal{A}^{(2)}$ | $\mathcal{A}^{(3)}$ | $\mathcal{A}^{(4)}$ | $\mathcal{A}^{(5)}$ |
|---|---|---|---|---|---|
| $\mathcal{A}^{(1)}$ | -14.4 | -15.8 | 1.5 | -5.4 | **10** |
| $\mathcal{A}^{(2)}$ | -13.2 | 5.8 | -8.7 | -2.2 | -18.2 |
| $\mathcal{A}^{(3)}$ | -5.9 | -19.0 | -0.7 | -2.0 | -19.5 |
| $\mathcal{A}^{(4)}$ | 0.8 | 4.7 | -14.8 | 2.5 | -4.1 |
| $\mathcal{A}^{(5)}$ | -11.3 | -8.2 | -20.0 | -17.3 | -17.6 |

Matrix 8

| $a_2$ \ $a_1$ | $\mathcal{A}^{(1)}$ | $\mathcal{A}^{(2)}$ | $\mathcal{A}^{(3)}$ | $\mathcal{A}^{(4)}$ | $\mathcal{A}^{(5)}$ |
|---|---|---|---|---|---|
| $\mathcal{A}^{(1)}$ | -1.4 | -19.2 | 7.2 | -5.5 | 7.4 |
| $\mathcal{A}^{(2)}$ | -18.5 | -20.0 | -14.4 | -17.6 | -5.1 |
| $\mathcal{A}^{(3)}$ | 3.6 | 5.5 | **10** | -13.3 | -4.9 |
| $\mathcal{A}^{(4)}$ | 9.8 | -12.3 | 0.6 | -16.5 | -13.0 |
| $\mathcal{A}^{(5)}$ | -11.8 | -20.0 | -2.4 | 7.1 | -2.3 |

Matrix 9

| $a_2$ \ $a_1$ | $\mathcal{A}^{(1)}$ | $\mathcal{A}^{(2)}$ | $\mathcal{A}^{(3)}$ | $\mathcal{A}^{(4)}$ | $\mathcal{A}^{(5)}$ |
|---|---|---|---|---|---|
| $\mathcal{A}^{(1)}$ | -4.5 | -5.2 | -8.4 | -8.9 | 5.5 |
| $\mathcal{A}^{(2)}$ | -12.4 | -9.5 | 8.8 | 5.4 | 4.4 |
| $\mathcal{A}^{(3)}$ | -4.6 | 1.3 | 5.5 | 7.3 | -6.8 |
| $\mathcal{A}^{(4)}$ | 9.0 | -18.7 | -18.2 | -13.7 | -8.2 |
| $\mathcal{A}^{(5)}$ | 2.2 | -9.1 | **10** | 7.1 | -20.0 |

Matrix 10

| $a_2$ \ $a_1$ | $\mathcal{A}^{(1)}$ | $\mathcal{A}^{(2)}$ | $\mathcal{A}^{(3)}$ | $\mathcal{A}^{(4)}$ | $\mathcal{A}^{(5)}$ |
|---|---|---|---|---|---|
| $\mathcal{A}^{(1)}$ | -8.4 | -1.8 | -20.0 | 7.3 | -3.0 |
| $\mathcal{A}^{(2)}$ | -8.7 | 1.7 | 4.8 | 2.0 | -7.8 |
| $\mathcal{A}^{(3)}$ | -13.3 | -3.2 | 0.7 | -1.8 | -10.7 |
| $\mathcal{A}^{(4)}$ | 9.8 | -12.3 | 0.6 | -16.5 | -13.0 |
| $\mathcal{A}^{(5)}$ | 1.8 | 2.9 | -1.1 | **10** | 8.2 |

## B.2 STARCRAFT II MICROMANAGEMENT (SMAC) TASKS

### B.2.1 BENCHMARKING ON STARCRAFT II MICROMANAGEMENT TASKS

In section 5.1, we have compared and discussed the advantage of our approach against baselines on several representative maps. Here we further compare our reproach against baselines on all maps. The SMAC benchmark contains 14 maps that have been classified as easy, hard, and super hard. In

this paper, we design one more map `3h_vs_1b1z3h`, whose difficulty is comparable with official super hard maps.

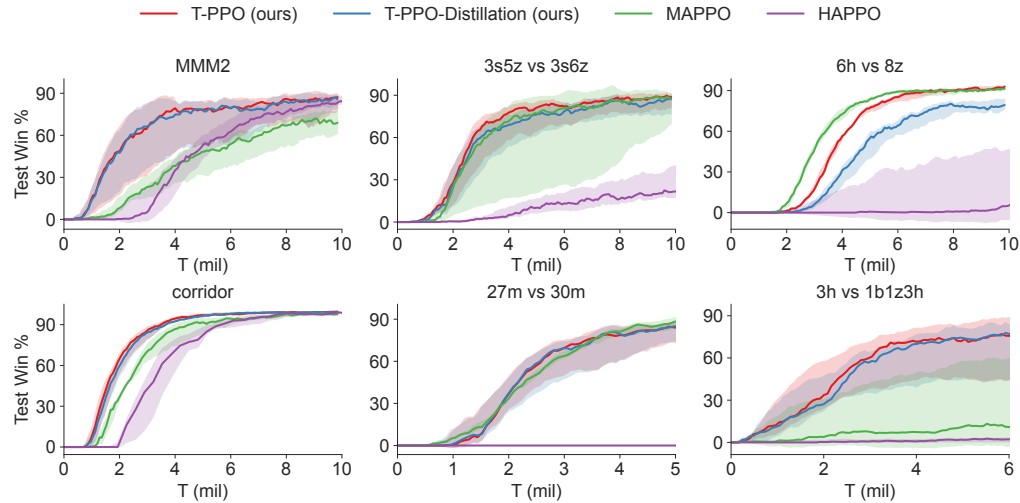

Figure 7: Comparisons between Our approach and policy-based baselines on all **superhard** maps.

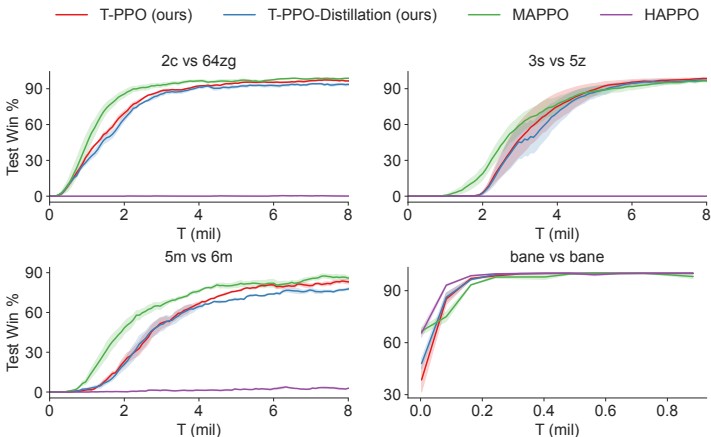

Figure 8: Comparisons between Our approach and policy-based baselines on all **hard** maps.

In Figure. 7, we compare the performance of our approach with baseline algorithms on all super hard maps. We can see that T-PPO outperforms all the baselines, especially on `3s5z_vs_3s6z`, `MMM2`, and `3h_vs_1b1z3h`. These results demonstrate that T-PPO can handle challenging tasks more efficiently with theoretical guarantees of its sequential transformation framework, in line with our expectations of it. Meanwhile, our distilled policy T-PPO-Distillation performs similarly to T-PPO, illustrating the competitiveness of our approach to fully decentralized evaluation. HAPPO performs poorly on 4 out of 6 super hard maps, demonstrating its limitation on complex tasks.

Our approach maintains its out-performance on most hard and easy maps. Compared with MAPPO, our approach achieves better convergence points on `3s_vs_5z` and `1c3s5z`, which confirms the experiment results and analysis in Section **??**. In summary, T-PPO establishes a new state of the art on SMAC benchmark by outperforming all policy-based baselines in 11 out of 15 scenarios. Meanwhile, the distilled strategy of T-PPO performs as well as the original strategy, which maintains a fairer comparison with baselines on fully decentralized execution.

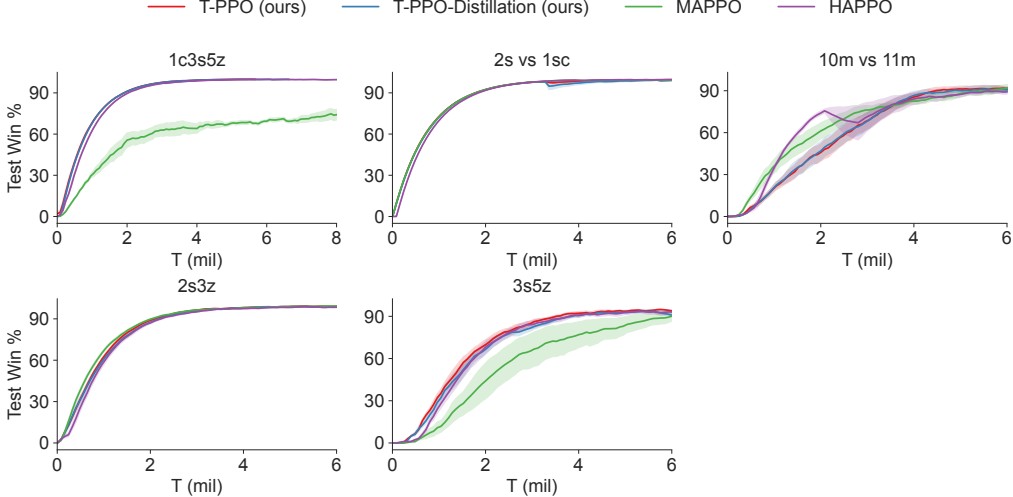

Figure 9: Comparisons between Our approach and policy-based baselines on all **easy** maps.

### B.2.2 HYPER-PARAMETERS

Our code is implemented based on MAPPO (https://github.com/marlbenchmark/on-policy). We share the same structure with MAPPO except improvement we mentioned in Section **??** to instantiate our transformation framework. Meanwhile, we share the same hyper-parameters with MAPPO (Yu et al., 2021) only except: (1) We fine-tune the weight of entropy on three maps (0.03 on `3h_vs_1b1z3h`, `6h_vs_8z`, and `5m_vs_6m`) for both our approach and MAPPO. (2) The hyper-parameters of multi-head attention (MHA) modules. As for HAPPO, we use the officially released code and related hyper-parameters (https://github.com/cyanrain7/TRPO-in-MARL).

Table 8: Common hyper-parameters for our approach in the SMAC domain.

| common hyperparameters | value |
|---|---|
| MHA heads | 3 |
| MHA latent dimension | 4 |
| MHA norm weight | 0.001 |
| entropy weight | 0.01 |

### B.2.3 DISTILLATION

The distillation is just independent behavioral cloning for each agent.

Denote the joint policy as $\pi_{\text{jt}}(\cdot|s)$, and the decentralized policy for agent $i$ as $\pi^i_{\text{decen}}(\cdot|s;\theta)$. The independent behavioral cloning is equivalent to minimization of the KL divergence between the joint policy $\pi_{\text{jt}}(\cdot|s)$ and the joint decentralized policy $\prod_{i=1}^n \pi^i_{\text{decen}}(\cdot|s;\theta)$.

$$
\text{KL}\left(\pi_{\text{jt}}(\cdot|s)\bigg\|\prod_{i=1}^n \pi^i_{\text{decen}}(\cdot|s;\theta)\right)
$$

$$
= \sum_{\boldsymbol{a}=(a_1,\cdots,a_n)} \pi_{\text{jt}}(\boldsymbol{a}|s)\log\frac{\pi_{\text{jt}}(\boldsymbol{a}|s)}{\prod_{i=1}^n \pi^i_{\text{decen}}(a_i|s;\theta)}
$$

$$
= \underbrace{\mathbb{E}_{\boldsymbol{a}\sim\pi_{\text{jt}}(\cdot|s)}\left[-\sum_{i=1}^n \log\pi^i_{\text{decen}}(a_i|s;\theta)\right]}_{\text{behavioural cloning loss}} - \underbrace{H(\pi_{\text{jt}}(\cdot|s))}_{\text{entropy (a constant)}}
$$

### B.3 GOOGLE RESEARCH FOOTBALL (GRF) TASKS

Because there are fewer agents in GRF compared with SMAC (3 in `Academy_3_vs_1_with_Keeper` and 4 in `Academy_Counterattack_Hard`, we slightly decrease the number of MHA heads from 3 to 2 as shown in Table 9. Other hyper-parameters remains the same as SMAC.

Table 9: Common hyper-parameters for our approach in the GRF domain.

| common hyperparameters | value |
|---|---|
| MHA heads | 2 |
| MHA latent dimension | 4 |
| MHA norm weight | 0.001 |
| entropy weight | 0.01 |

### B.4 RUNNING TIME OF T-PPO

The sequential update does take longer time than the concurrent update in the training phase, while in the testing phase, our algorithm doesn't take extra time since a decentralized policy is already calculated by distillation.

Specifically, in the training phase, our framework takes $n$ times the time to do action inference, where $n$ is the number of agents. However, it's worth mentioning that, the time of action inference is only part of the time doing a whole training iteration, which also includes environment simulation and policy training. On the whole, the training time cost of our framework is 0.91 times more than MAPPO in SMAC environment `3m` (3 agents), and 1.76 times more in SMAC environment `10m vs 11m` (10 agents). This result embodies a trade-off between training time and training performance.

Specific time of each part is shown Table 10.

Table 10: Comparison between TPPO and MAPPO on training time

| Env: 3m (1M) | Action inference: $t_I$ | Env simulation: $t_S$ | Env interaction: $t_I + t_S$ | Policy Training: $t_P$ | The whole training phase: $t_I + t_S + t_P$ |
|---|---|---|---|---|---|
| T-PPO | 1694.6s | 915.4s | 2610s | 233.7s | 2843.7s |
| MAPPO | 434.9s | 904.5s | 1339.4s | 151.9s | 1491.3s |
| Ratio | 3.9 | 1.01 | 1.95 | 1.54 | 1.91 |
| Env: 10m vs 11m (1M) | Action inference: $t_I$ | Env simulation: $t_S$ | Env interaction: $t_I + t_S$ | Policy Training: $t_P$ | The whole training phase: $t_I + t_S + t_P$ |
| T-PPO | 5011.7s | 2065.5s | 7077.2s | 794.1s | 7871.1s |
| MAPPO | 501.1s | 2063.2s | 2564.3s | 292.1s | 2856.4s |
| Ratio | 10.0 | 1.0 | 2.76 | 2.72 | 2.76 |

### B.5 COMPARISON BETWEEN T-PPO AND OTHER ACTOR-CRITIC BASELINES

We also compare T-PPO with two more actor-critic methods – DOP and FOP on three superhard SMAC maps for completeness. The result is shown in Figure 10.

### B.6 T-DQN RESULTS

We also implement the off-policy method T-DQN based on the single agent algorithm DQN (Mnih et al., 2013) for completeness. In Figure 11 We evaluate DQN on a Markov Game (Matrix games with random transition), a **hard** and a **superhard** SMAC maps to show that our framework is also compatible with off-policy methods.

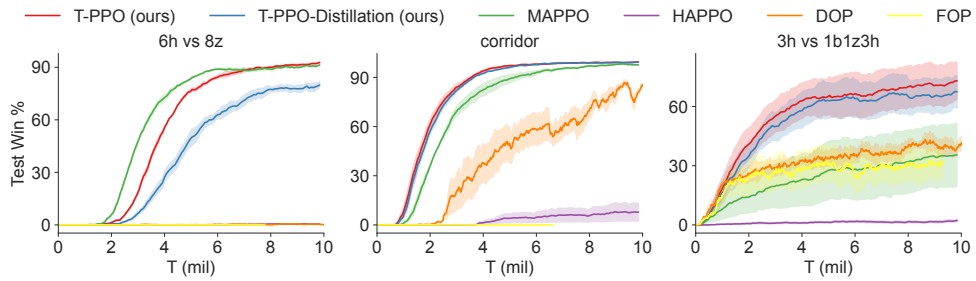

Figure 10: Comparisons between T-PPO and other on-policy baselines on three **superhard** maps.

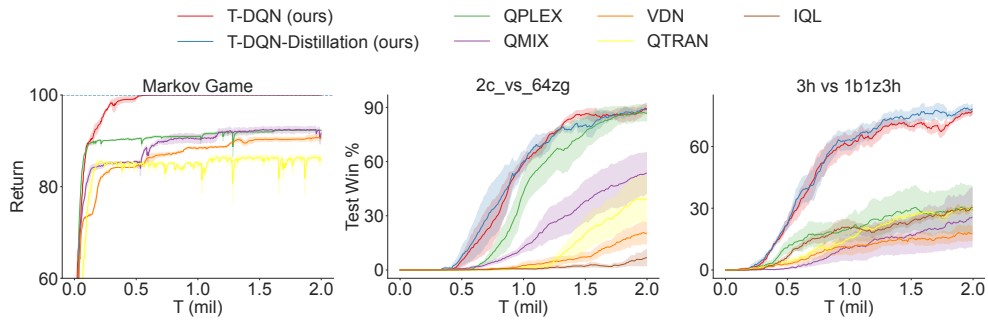

Figure 11: Comparisons between T-DQN and off-policy baselines on Markov Game, a **hard** and a **superhard** SMAC maps.

### B.7 LEARNED BEHAVIOUR OF THE SEQUENTIAL FRAMEWORK

We visualize the policy learned by our approach and compare it with MAPPO in MMM2. Based on the comparison, we notice an interesting phenomenon. The joint strategy trained by MAPPO is usually conservative, only moving in a small area, and only two agents are left in the end. On the contrary, the joint strategy trained by our approach is more aggressive. Our agents pull back and forth frequently based on opponents' movement in a large-scale range while ensuring effective fire focus. We think this phenomenon is caused by our sequential transformation framework, which enables each agent to fully understand the team strategy for more efficient coordination.

