# OpenReview forum: "Towards Global Optimality in Cooperative MARL with Sequential Transformation"
_ICLR.cc/2023/Conference — Submitted to ICLR 2023_

### Official Review · Reviewer_toNr · 2022-10-24

**Confidence:** 4
**Correctness:** 2
**Technical Novelty And Significance:** 2
**Empirical Novelty And Significance:** 2
**Recommendation:** 1

**Clarity, Quality, Novelty And Reproducibility:**

- Clarity: I can roughly understand the main idea of this paper, however, some points are confusing. (See Strength And Weaknesses)
- Quality:  It seems a hastily written paper.
- Novelty: The perspective of transformation is novel. But the underlying methodology is not surprising since it seems a weak version of MARL communication or agent modeling.
- Reproducibility: the authors do not provide the source code.


**Strength And Weaknesses:**

Pros:

- This paper takes on an interesting problem in MARL: the improvement of the CTDE scheme outside value decomposition and centralized critic.
- The experiments are completed.

However, I have several concerns here:
- The paper is not well-written and not well-organized with many typos and confusing notation. It seems a hastily written paper.
- The description of Theorem 3.1 is wrong, single step MDP is not MMDP. The proof of Theorem 3.1 is confusing. The example game is a pure coordination game, which has |A| pure strategy equilibria. However, it is the nature of the pure coordination game, which does not mean that multi-agent actor-critic algorithms create inherent local minimums.
- The reason why the proposed method works in “Multi-task Matrix Games” as shown in Fig.1 is the agents are factually communicating to avoid miscoordination in such one-shot games You can check examples 3.2 and 3.3 in [1].
- The order of actions matters.
- If agent $i$ can know all previous agents’ actions, this case is a weak version of MARL communication or agent modeling.
- The distillation is missing in the main part of this paper.

Minor:

- Reference in line 3 of page 3: missing parentheses
- see eq. (1)/appendix/section/theorem/proposition/definition -> see Eq.1/Appendix/Section/Theorem/Proposition/Definition
- $\Gamma M$ in the last line of page 5: missing parentheses
- Proposition 4.1 (suitable …) -> Proposition 4.1 (Suitable …)
- In Prop. 4.1, “if Assumption 4.1, 4.2, 4.3 in Liu et al. (2019) holds” -> “if Assumption 4.1, 4.2, and 4.3 in Liu et al. (2019) hold”
- So many typos, grammar mistakes, and irregular notations!!!


[1] When Does Evolution Lead to Efficiency in Communication Games. 1994.


**Summary Of The Paper:**

This paper mainly considers the transformation from centralized training with decentralized execution (CTDE) in multi-agent reinforcement learning (MARL) to single-agent reinforcement learning (SARL). Specifically, the authors reformulate a multi-agent MDP as a special "single-agent" MDP with a sequential decision-making structure among agents. In this way, SARL algorithms can be used during training, and a distilled policy is used during execution. Experiments show a good performance in a few scenarios in SMAC and GRF.

**Summary Of The Review:**

I think the authors should improve this paper with much effort. The current version is not satisfying in terms of both presentation and the core idea.

---

> ### Author Response · Authors · 2022-11-18
> **Reply to toNr**
>
> Thank you for your comprehensive comments. We have updated the submission to fix typos.
>
> **Q3.1** In Theorem 3.1, single step MDP is not MMDP
>
> **A3.1**
>
> We have fixed the typo.
>
>
>
> **Q3.2** The proof of Theorem 3.1 is confusing.
>
> **A3.2**
>
> It's not trivial for an NE in a cooperative game to serve a local optimum in the multi-agent actor loss. The reviewer may read the proof again, we believe it is clear enough.
>
>
>
> **Q3.3** The order of actions matters.
>
> **A3.3**
>
> According to the definition and analysis of sequential transformation in section 4.1, any fixed order would suffice for all claimed properties in the paper. The further discussion of sequence ordering beyonds the scope of this paper and can serve as future work.
>
> **Q3.4**
>
> If agent $i$ can know all previous agents’ actions, this case is a weak version of MARL communication or agent modeling.
>
> **A3.4**
>
> We emphasize here that our T-PPO-Distillation has a decentralized policy and requires no previous actions during execution.
>
> **Q3.5**
>
> The distillation is missing in the main part of this paper.
>
> **A3.5**
>
> We did explain the distillation at the end of Section 4.2 and Appendix B.2.3.

---

### Official Review · Reviewer_SxZS · 2022-10-25

**Confidence:** 3
**Correctness:** 2
**Technical Novelty And Significance:** 4
**Empirical Novelty And Significance:** 2
**Recommendation:** 3

**Clarity, Quality, Novelty And Reproducibility:**

### Clarity
* Existing issues are well explained.
* Proposed method could use more explaining
* ‘In the environment setting, we use sparse rewards with both SCORING and CHECKPOINT for our approach and all baselines. For observations, we follow (Li et al., 2021a), using simple 115 as the observation while removing the information irrelevant to the current scenario.” -> what does 115 mean here?

### Quality
* I found the use of distillation poorly motivated could more information be provided to explain how this process works for each agent (are the distillations conditioned on the specific agent or are they n copies)

### Novelty
* Transformation is novel for CTDE


**Strength And Weaknesses:**

### Strengths

Strong theoretical work to show existing models are more susceptible to falling into local optima than the proposed method.

### Weaknesses

* Unclear why they only evaluate on 6 of the SMAC environments, there are many more.
* They have not used any SMAC maps with > 10 agents
* Do they provide other agents’ actions at test time (as suggested by Figure 2)? If so this makes this an entirely unfair comparison to CTDE.
* I would like to understand the importance of the ordering of agents within the “virtual” episode.
* I would like ablations to understand the importance of the MHA and KLD terms.
* The MAPPO results are much lower than reported in the original paper - can you explain the performance difference


**Summary Of The Paper:**

The authors propose a revision of the existing approach to Centralised Training Decentralised Executation (CTDE)  for the Cooperative AI problems. They propose a transformation from the MMDP to a MDP problem, thus allowing them to use Single Agent RL algorithms. This is motivated by observations (with theoretical justification) that current CTDE approaches with gradient based learners introduce a large set of local equilibria which restrict the ability to converge to optimal behaviour.

The authors then implement a simple of this transform (which amounts to transforming agents from taking simultaneous moves into sequential moves and using a single agent instead of a team). Implementation details are provided such as using Multi-Head Attention, KL Divergence and Distillation (although not all components are ablated).

Performance is measured on saturated benchmarks (SMAC) and so performance can only been seen to be a marginal improvement.


**Summary Of The Review:**

Paper exposes a theoretical problem with CTDE which is a good contribution.

The solution proposes lacks motivation or compelling results to be taken seriously.

---

> ### Author Response · Authors · 2022-11-18
> **Reply to SxZS**
>
> Thank you for your comments. We provide clarification to your questions below.
>
> **Q2.1**
>
> Unclear why they only evaluate on 6 of the SMAC environments, there are many more.
>
> **A2.1**
>
> We have evaluated 6 super-hard, 4 hard, and 5 easy maps of SMAC environment. We put the 6 of them in the main body and the rest in Appendix B.2.1 due to the lack of space.
>
> We have tested our algorithm in SMAC, GRF, and our didactic toy example. There are a lot of figures, so it's impracticable to put all of them in the main body. We are sorry for not mentioning it in the main body.
>
> **Q2.2**
>
> They have not used any SMAC maps with > 10 agents
>
> **A2.2**
>
> We do have used maps with > 10 agents, like 10m_vs_11m and 27m_vs_30m. One can find it in Appendix B.2.1.
>
> **Q2.3**
>
> Do they provide other agents’ actions at test time (as suggested by Figure 2)? If so this makes this an entirely unfair comparison to CTDE.
>
> **A2.4**
>
> We have plotted both communication policy (T-PPO) and decentralized policy (T-PPO-Distillation) in each figure. In T-PPO-Distillation, other agents' actions are not provided.
>
> **Q2.5**
>
> I would like to understand the importance of the ordering of agents within the “virtual” episode.
>
> **A2.5**
>
> According to the definition and analysis of sequential transformation in section 4.1, any fixed order would suffice for all claimed properties in the paper. The further discussion of sequence ordering beyonds the scope of this paper and can serve as future work.
>
> **Q2.6**
>
> I would like ablations to understand the importance of the MHA and KLD terms.
>
> **A2.6**
>
> We try to ablate our MHA module in two classical maps, 3h vs 1b1z3h and MMM2. We find the average winning rate will decrease to zero after the ablation, indicating the significance of our multi-head attention module in information extraction.
>
> **Q2.7**
>
> The MAPPO results are much lower than reported in the original paper - can you explain the performance difference
>
> **A2.7**
>
> MAPPO uses different hyper-parameters and engineering tricks for each different map. For example, stacked frames as observation only in 3s vs 5z, which increase the performance a lot. For a fairer comparison with current MARL algorithms, we mute a few tricks that are not commonly used, which may lead to a decrease in performance. T-PPO is implemented with the same hyper-parameters as MAPPO. They only differ in network architecture and training manner. Therefore our comparison between T-PPO and MAPPO is fair.
>
> **Q2.8**
>
> ‘In the environment setting, we use sparse rewards with both SCORING and CHECKPOINT for our approach and all baselines. For observations, we follow (Li et al., 2021a), using simple 115 as the observation while removing the information irrelevant to the current scenario.” -> what does 115 mean here?
>
> **A2.8**
>
> It is an officially provided option of observation parameterization: a 115-dimensional float vector. See [1] for clarity. We have fixed the typo in the main body.
>
> [1] Karol Kurach, Anton Raichuk, Piotr Sta ́nczyk, Michał Zaj  ̨ac, Olivier Bachem, Lasse Espeholt, Car-
> los Riquelme, Damien Vincent, Marcin Michalski, Olivier Bousquet, et al. Google research
> football: A novel reinforcement learning environment. arXiv preprint arXiv:1907.11180, 2019.
>
> **Q2.9**
>
> Are the distillations conditioned on the specific agent or are they n copies?
>
> **A2.9**
>
> The distillation part is illustrated in Figure 2. And as mentioned in Section 4.2, we share GRU and the representation layer, i.e. conditioned on the specific agent.

---

### Official Review · Reviewer_fgdt · 2022-11-04

**Confidence:** 2
**Correctness:** 2
**Technical Novelty And Significance:** 2
**Empirical Novelty And Significance:** 3
**Recommendation:** 3

**Clarity, Quality, Novelty And Reproducibility:**

As discussed before, the idea of this paper is considered to be novel. However, the writing clarity and quality need some improvement. In particular, the definitions need to be made precise, the arguments need to be made rigorous and the missing details need to be filled.

### Questions
- Does the problem of many local minimums still exist for the reduced single-agent RL problem?

**Strength And Weaknesses:**

### Strengths
The reduction from cooperative MARL problem to SARL problem is considered to be novel. Meanwhile, the experiment results also look promising.

### Weaknesses
One of the major weakenesses of this paper lies in its theoretical support. In particular, although the problem of multi-agent games may contain many local minimums, it is not clear how this problem is resolved in the transformed MDP. Since the problem is in general non-convex, many local minimums may still exist in the transformed MDP. It will be better if it can provide an analysis that the proposed framework does resolve the problem of many local minimums even in some toy examples.

Meanwhile, the proofs provided in Appendix seem to miss a lot of details, together with some hand-waving parts. For example, in the proof of Proposition A.1:
- It will be better to state clear how the formulas from QPLEX are simplified.
- In the first equation, $\theta_1, \dots, \theta_n$ does not appear in the right-hand side.
- The loss function $C$ is not clearly defined.
- It is not clear what "the best we can expect" means.
- The proposition claims to find some $\phi$ such that $(\theta_1, \dots, \theta_n, \phi)$ is a local minimum of $C$, but the proof does not make clear how exactly $\phi$ looks like.
- It will also be better to add the omiited details of Theorem 3.2 back.

These hand-waving arguments and missing details make the theoretical claims in this paper less convincing.

**Summary Of The Paper:**

This paper studies centralized training with decentralized execution (CTDE) in cooperative multi-agent reinforcement learning (MARL). It first analyzes disadvantages of previous algorithms from an optimization perspective. Then, it proposes a framework that reduces a cooperative MARL problem to a single-agent RL problem and a corresponding algorithm called transformed PPO (T-PPO). This paper also presents experiment results on simple matrix games, Starcraft II and Google Research Football and show superior performance of the proposed T-PPO algorithm.

**Summary Of The Review:**

This paper proposes a novel reduction from cooperative MARL problem to SARL problem and the correspondinng experiment results look promising. However, its current theoretical support is not solid enough and its writing also needs some improvement.

---

> ### Author Response · Authors · 2022-11-18
> **Reply to fgdt**
>
>
> Thank you for your comments. We provide clarification to your questions below.
>
> **Q1.1**
>
> It is not clear how this problem is resolved in the transformed MDP. Since the problem is in general non-convex, many local minimums may still exist in the transformed MDP.
>
> **A1.1**
>
> The structure of the loss function in the transformed MDP depends on the single-agent algorithm you choose. Theorem 4.1 guarantees that T-A learns the optimal policy on MMDP if A has an optimal guarantee on MDP. There are a lot of single-agent algorithms with optimality guarantee, e.g.
>
> - a variety of tabular learning algorithms like value iteration and Q-learning.
> - some algorithms with neural networks, as mentioned in Proposition 4.1 -- some suitable implementations of PPO (see [1] for detail).
>
> [1] Boyi Liu, Qi Cai, Zhuoran Yang, and Zhaoran Wang. Neural proximal/trust region policy optimization attains globally optimal policy, 2019. URL https://arxiv.org/abs/1906.10306.
>
> **Q1.2**
>
> The proofs provided in Appendix seem to miss a lot of details.
>
> **A1.2**
>
> We are sorry for our previously oversimplified proof. We have updated rigorous proofs for proposition A.1 and Theorem 3.2. Hope this would address your concern.
>
> **Q1.3**
>
> Does the problem of many local minimums still exist for the reduced single-agent RL problem?
>
> **A1.3**
>
> As mentioned in A1.1, the structure of the loss function in the transformed MDP depends on the single-agent algorithm you choose.

---

> > ### Comment · Reviewer_fgdt · 2022-11-28
> > **Response**
> >
> > Thank you very much for your update! However, from my perspective, the updated proof is still hard to understand and thus I'll keep my score. It's probably better to further improve the writing for the next submission.

---

### Decision · Program_Chairs · 2023-01-20

**Decision:**

Reject

**Justification For Why Not Higher Score:**

Concerns about the optimization formulation, the proofs, the experiments, and the writing.

**Justification For Why Not Lower Score:**

N/A

**Metareview: Summary, Strengths And Weaknesses:**

This paper studies centralized training with decentralized execution MARL It first analyzes the disadvantages of previous algorithms from an optimization perspective. Then, it proposes a framework that reduces a cooperative MARL problem to a single-agent RL problem. The reviewers raised concerns about the optimization formulation, the proofs, the experiments, and the writing. The AC agrees and recommends rejection.